# Coexistence of Dominant Marine Phytoplankton Sustained by Nutrient Specialization

Takako Masuda,[a,b]* Keisuke Inomura,[c] Jan Mareš,[b,d,e] Taketoshi Kodama,[a] Takuhei Shiozaki,[a]§ Takato Matsui,[f] Koji Suzuki,[f] Shigenobu Takeda,[a]◇ Curtis Deutsch,[g] Ondřej Prášil,[b] Ken Furuya[a]∞

[a]Department of Aquatic Bioscience, The University of Tokyo, Bunkyo, Tokyo, Japan

[b]Institute of Microbiology, The Czech Academy of Sciences, Třeboň, Czechia

[c]Graduate School of Oceanography, University of Rhode Island, Narragansett, Rhode Island, USA

[d]Institute of Hydrobiology, Biology Centre of the Czech Academy of Sciences, České Budejovice, Czechia

[e]Department of Botany, University of South Bohemia, Faculty of Science, České Budejovice, Czechia

[f]Graduate School of Environmental Science/Faculty of Environmental Earth Science, Hokkaido University, Sapporo, Japan

[g]Department of Geosciences, Princeton University, Princeton, New Jersey, USA

Takako Masuda and Keisuke Inomura contributed equally to the article. The order of the two authors was determined based on the chronology of the study.

**ABSTRACT** *Prochlorococcus* and *Synechococcus* are the two dominant picocyanobacteria in the low-nutrient surface waters of the subtropical ocean, but the basis for their coexistence has not been quantitatively demonstrated. Here, we combine *in situ* microcosm experiments and an ecological model to show that this coexistence can be sustained by specialization in the uptake of distinct nitrogen (N) substrates at low-level concentrations that prevail in subtropical environments. In field incubations, the response of both *Prochlorococcus* and *Synechococcus* to nanomolar N amendments demonstrates N limitation of growth in both populations. However, *Prochlorococcus* showed a higher affinity to ammonium, whereas *Synechococcus* was more adapted to nitrate uptake. A simple ecological model demonstrates that the differential nutrient preference inferred from field experiments with these genera may sustain their coexistence. It also predicts that as the supply of $NO_3^-$ decreases, as expected under climate warming, the dominant genera should undergo a nonlinear shift from *Synechococcus* to *Prochlorococcus,* a pattern that is supported by subtropical field observations. Our study suggests that the evolution of differential nutrient affinities is an important mechanism for sustaining the coexistence of genera and that climate change is likely to shift the relative abundance of the dominant plankton genera in the largest biomes in the ocean.

**IMPORTANCE** Our manuscript addresses the following fundamental question in microbial ecology: how do different plankton using the same essential nutrients coexist? *Prochlorococcus* and *Synechococcus* are the two dominant picocyanobacteria in the low-nutrient surface waters of the subtropical ocean, which support a significant amount of marine primary production. The geographical distributions of these two organisms are largely overlapping, but the basis for their coexistence in these biomes remains unclear. In this study, we combined *in situ* microcosm experiments and an ecosystem model to show that the coexistence of these two organisms can arise from specialization in the uptake of distinct nitrogen substrates; *Prochlorococcus* prefers ammonium, whereas *Synechococcus* prefers nitrate when these nutrients exist at low concentrations. Our framework can be used for simulating and predicting the coexistence in the future ocean and may provide hints toward understanding other similar types of coexistence.

**KEYWORDS** nutrient specialization, *Prochlorococcus*, *Synechococcus*

Address correspondence to Takako Masuda, takakom@affrc.go.jp.

*Present address: Takako Masuda, Fisheries Resources Institute, Japan Fisheries Research and Education Agency, Shiogama, Miyagi, Japan.

§Present address: Takuhei Shiozaki, Atmosphere and Ocean Research Institute, The University of Tokyo, Kashiwa, Chiba, Japan.

◇Present address: Shigenobu Takeda, Graduate School of Fisheries and Environmental Sciences, Nagasaki University, Bunkyo, Nagasaki, Japan.

∞Present address: Ken Furuya, Graduate School of Engineering, Soka University, Hachioji, Tokyo, Japan.

The authors declare no conflict of interest.

Marine phytoplankton are key drivers of the global carbon cycle (1, 2), photosynthesizing about $50 \times 10^{15}$ g carbon (C) annually, which is comparable to that of the global terrestrial biosphere (1). Nearly a quarter of this net primary production can be accounted for by the two marine cyanobacteria *Prochlorococcus* and *Synechococcus* (3). The geographical distributions of the two groups are largely overlapping in the latitude band from ~40°N to ~40°S (4). Because of their dominance in such vast oceanic regions, understanding their niche partitioning is critical in predicting the future ecosystem and biogeochemical cycling. However, how these organisms coexist has still been in question.

Nutrient utilization is a critical factor in microbial coexistence (5–7). In the stratified open ocean, nutrient supply is weak, and concentrations are commonly drawn down to nanomolar levels that limit the rates of growth and uptake by phytoplankton (8). In these biomes, the mechanisms by which phytoplankton species partition scarce resources determine the relative abundance of species and thus community composition. However, a long-standing theory predicts that the species whose nutrient acquisition saturates at the lowest concentration will drive other organisms to extinction (9), such that coexistence requires specialization on distinct nutrient resources. Qualitative evidence supports differential nitrogen utilization in *Prochlorococcus* and *Synechococcus* (10), but the traits and conditions that favor the coexistence or dominance of these genera have not yet been quantitatively demonstrated. For example, models that assume similar nitrogen affinities for *Prochlorococcus* and *Synechococcus* (11) promote competitive exclusion, even where coexistence is widely observed (e.g., in the North Atlantic Gyre) (3).

To quantify the causes of niche separation and coexistence of *Prochlorococcus* and *Synechococcus*, we conducted a suite of nutrient enrichments at a nanomolar level to simulate realistic perturbations at a fixed station in the North Pacific (12°N, 135°E). The North Pacific subtropical gyre, where *Prochlorococcus* and *Synechococcus* are numerically predominant in the phytoplankton community, is characterized by extremely low surface nutrient concentrations at the nanomolar level with temporal and spatial variation (6, 12–15). We analyzed the data on growth and nutrient uptake using a simple population model for each species and applied the model to determine whether the coexistence of these species can be maintained.

## RESULTS AND DISCUSSION

***In situ* experiments.** To test the effect of increased nutrients at the nanomolar level, we conducted five nitrogen (N) addition experiments (M1 to M5) at the surface of a station in the subtropical Northwestern Pacific (12°N, 135°E) from 6 to 25 June 2008 during the MR08-02 cruise aboard the R/V *MIRAI*. Grazing was eliminated by filtration (see details in Materials and Methods) so that changes in phytoplankton abundance could be considered cell growth or cell death. To verify that only N was limiting, we also conducted three Fe-addition (Fe1 to Fe3) experiments (Table S8 in reference 16). Neither P nor Fe enrichment increased the abundance of any of the tested phytoplankton groups, including *Prochlorococcus* and *Synechococcus* (Fig. 1; see Fig. S1A and B and Table S1 in the supplemental material). Initial nutrient concentrations remained low during the cruise (<45 nM for N, ~30 nM for P, and more than 0.1 nM for dissolved iron) (see Table S2 in the supplemental material).

With the N addition, the growth of both *Prochlorococcus* and *Synechococcus* was stimulated regardless of its chemical form (repeated measures analysis of variance [RM-ANOVA], $\alpha < 0.05$) (Fig. 1; Fig. S1C and D; Table S1; see Table S3 in the supplemental material). However, the magnitude of growth responses differed between the oxidized and reduced forms of N, and these differences were genus specific (Fig. 1; Fig. S1C and D; Table S1 and S3). The growth of *Prochlorococcus* was more stimulated by ammonium ($NH_4^+$), while *Synechococcus* responded more strongly to nitrate ($NO_3^-$) (Fig. 1; Table S3). The enriched $NH_4^+$ was depleted on the third day, leading to decreased populations in both organisms (Fig. 1; see Fig. S2 in the supplemental

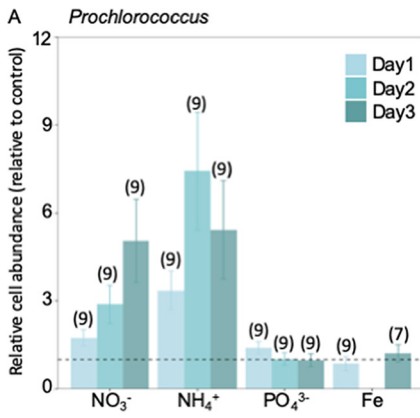
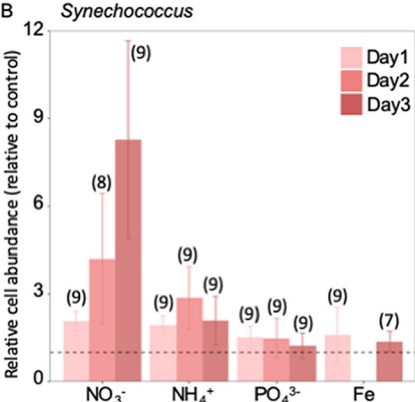

**FIG 1** The effects of different chemical forms of N, P, and Fe additions on cell abundances of *Prochlorococcus* (A) and *Synechococcus* (B) relative to controls (means ± SD), with the number of samples in parentheses. For N and P addition, data are based on experiments M1 to M3, where the initial concentrations of N were small (<10 nM) (Table S2). It shows that *Prochlorococcus* prefers $NH_4^+$ while *Synechococcus* prefers $NO_3^-$. Note that added $NH_4^+$ was depleted on the third day, leading to decreased populations in both organisms (Fig. S2). The dotted line at 1 represents no change from the control.

material). Urea stimulated the growth of *Prochlorococcus* and *Synechococcus* similar to or sometimes better than ammonium or nitrate, respectively (Fig. S1C and D; Table S3).

*Prochlorococcus* showed a significantly higher peak abundance in the $NH_4^+$ additions relative to the $NO_3^-$ additions in all five N and P bioassays (RM-ANOVA, $P < 0.05$) (Fig. 1; Fig. S1C; Table S3), indicating a higher affinity to $NH_4^+$ than to $NO_3^-$. This affinity difference can be explained by the low electron requirement to assimilate $NH_4^+$, which has been shown previously in *Prochlorococcus* both in culture and *in situ* (17–19).

Unlike *Prochlorococcus*, *Synechococcus* increased in the $NO_3^-$ amendment significantly more than in any other N treatments at the nanomolar level (ANOVA, $\alpha < 0.05$) (Fig. 1B: Fig. S1D; Table S3). The preference for $NO_3^-$ over $NH_4^+$ among *Synechococcus* has been consistent with results of the previous *in situ* bioassay experiments in the oligotrophic North Pacific Subtropical Gyre (17). A preference for $NO_3^-$ over $NH_4^+$ among *Synechococcus* seems at odds with both the energetic advantage of reduced N and with the direct observations in pure culture, which showed faster growth on $NH_4^+$ when $NH_4^+$ is relatively abundant (19, 20). The following modeling exercise reconciles these apparently conflicting observations, based on the differentiated half-saturation constants and affinity for $NO_3^-$ and $NH_4^+$.

**Ecological model for N uptake preferences.** To discern the mechanisms that sustain the coexistence of *Prochlorococcus* and *Synechococcus*, we used a simple ecological model (6, 7, 21, 22). The model represents distinct physiological traits for growth and nutrient affinity for each population and allows for their ecological interaction for the following two different N resources: $NH_4^+$ and $NO_3^-$. We have selected $NO_3^-$ and $NH_4^+$ to be part of the model mainly to focus on delivering the most striking finding extracted from the data—the different affinity to chemical forms of inorganic nitrogen between two organisms. Inorganic nutrients are also more commonly used in ecological modeling, and we have followed the previous similar studies focusing on inorganic nutrients (6, 7).

Despite its simplicity, the model reproduces the observed trends in populations over the course of the field study (Fig. 2; Fig. S3). The key parameters of the model were half-saturation constants ($K$) (see Table S4 in the supplemental material). $K$ values have been thought to represent the density of transporters on the cellular surface (23–25), with a lower value representing a higher transporter density. A lower $K_{NH4}$ for *Prochlorococcus* was essential to reproduce the rapid *Prochlorococcus* growth up to day 2 and the slower *Synechococcus* growth with $NH_4^+$ addition (Fig. 2A); when the species' relative $NH_4^+$ affinities are reversed, the model results deviate from the data (see Fig. S4 in the supplemental material). On the other hand, $K$ for $NO_3^-$ ($K_{NO3}$) must be lower

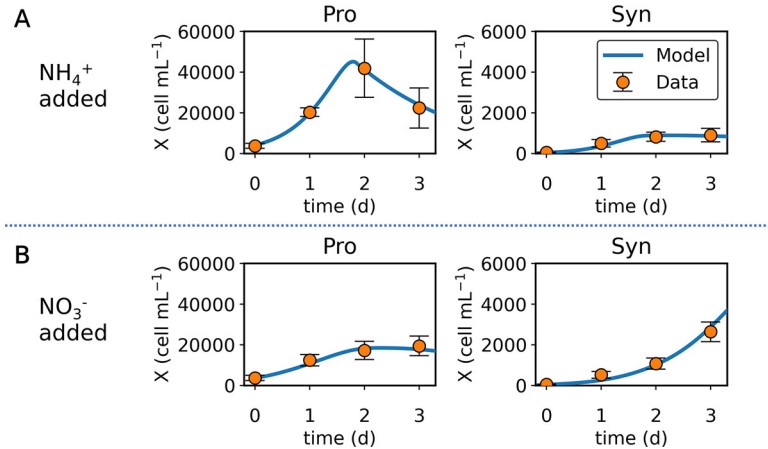

**FIG 2** Model data comparison of the time series of the abundance (*X*) of *Prochlorococcus* and *Synechococcus*. (A) $NH_4^+$ added. (B) $NO_3^-$ added. Points, data; curves, model; Pro, *Prochlorococcus*; Syn, *Synechococcus*. Data are based on experiments M1 to M3, where the initial concentrations of N were small (<10 nM) (Table S2). The error bars represent the standard deviation based on the mean values across from the selected experiments.

for *Synechococcus* to reproduce its continued high growth up to day 3 (Fig. 2B), and a reversal of species $NO_3^-$ affinities again results in a poor model fit to the data (see Fig. S5 in the supplemental material). Decreased $K_{NO3}$ of *Prochlorococcus* also causes model results to diverge from the data (compare Fig. 2 and Fig. S5).

The model parametrization suggests a higher maximum uptake ($V_{max}$) of $NO_3^-$ by *Synechococcus* than that of *Prochlorococcus* (Table S4), which also may contribute to the niche differentiation. It also indicates that $V_{max}$ for $NH_4^+$ is higher than that for $NO_3^-$ both for *Prochlorococcus* and *Synechococcus*. Thus, under high nutrient concentrations, *Synechococcus* may grow faster with $NH_4^+$ than with $NO_3^-$ (Table 1; Fig. 3). This model implication is consistent with the previous experiments where a large amount (micromolar level) of $NH_4^+$ and $NO_3^-$ is added (19, 20). However, the affinity ($A = V_{max}/K$) for $NO_3^-$ turned out to be higher than that for $NH_4^+$ for *Synechococcus* (see Table S5 in the supplemental material). The affinity provides an alternative measure of the relative ability of various species to compete for nutrients (24, 26). Because the affinity is the initial slope of nutrient uptake versus nutrient concentration relationship (27), the value of affinity is especially relevant when nutrients are depleted as in the subtropical gyres, where small phytoplankton tend to dominate (28). The result shows *Synechococcus* has an advantage for $NO_3^-$ and *Prochlorococcus* has an advantage for $NH_4^+$ (Table S5), consistent with what is suggested by the predicted half-saturation values (and not by maximum uptake rates). These results collectively suggest that in a low-nutrient environment as in the subtropical gyres, these organisms have different nitrogen uptake preferences for $NH_4^+$ and $NO_3^-$ (Table 1; Fig. 3), allowing their coexistence, and the differentiated half-saturation constants are the key contributors.

To test the effect of different affinity and *K* values on the coexistence of *Prochlorococcus* and *Synechococcus*, we obtained a steady-state solution for cellular nitrogen per volume of water for various ratios of $K_{NO3}^{Pro}/K_{NO3}^{Syn}$ and $K_{NH4}^{Pro}/K_{NH4}^{Syn}$ (note that these relationships are the inversely

**TABLE 1** Preferred nutrient for *Prochlococcus* and *Synechococcus* under different nutrient concentrations

|  | Preferred nutrient under: | |
| --- | --- | --- |
| **Genus** | **Low-nutrient concentration**[a] | **High-nutrient concentration**[a] |
| *Prochlorococcus* | $NH_4^+$ | $NH_4^+$ |
| *Synechococcus* | $NO_3^-$ | $NH_4^+$ |

[a]Broadly, low nutrient indicates lower nanomolar level (e.g., surface of subtropical gyres [e.g., this study]) and high nutrient indicates above that level (e.g., initial conditions of batch culture experiments [19, 20, 96]). Specifically, the nutrient preference of *Synechococcus* is predicted to be flipped at 39 nM (Fig. 3).

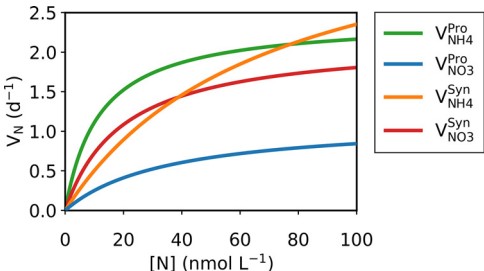

**FIG 3** Nutrient uptake rate $V_N$ against nutrient concentration [*N*]. In the legend, *Pro* and *Syn* indicate *Prochlorococcus* and *Synechococcus*, respectively, and *NH4* and *NO3* indicate ammonium and nitrate, resepectively. For example, $V_{NH4}^{Pro}$ indicates ammonium uptake rate by *Prochlorococcus* versus ammonium concentrations.

proportional to the affinity ratios given the constant $V_{max}$ values). The result shows that these *K* value ratios strongly influence the coexistence of these organisms at given resource ratios of $NO_3^-$ and $NH_4^+$. For example, when the resource ratio of $NO_3^-$ and $NH_4^+$ is 1:3, the model shows that the set of *K* values that reproduce the observations (Fig. 1, 2) allows coexistence with a similar amount of biomass, where *Prochlorococcus* uses mainly $NH_4^+$ and *Synechococcus* uses mainly $NO_3^-$ (see the cyan triangle in Fig. 4A and the domain explanation in Fig. 4B, zone i). If $K_{NO3}^{Pro}/K_{NO3}^{Syn}$ was lower, we would expect to see only *Prochlorococcus* (Fig. 4B, zone iii); however, if $K_{NH4}^{Pro}/K_{NH4}^{Syn}$ was sufficiently higher, *Synechococcus* would exclude *Prochlorococcus* by outcompeting them for $NH_4^+$ (Fig. 4B, zone ii). If both $K_{NO3}^{Pro}/K_{NO3}^{Syn}$ was lower and $K_{NH4}^{Pro}/K_{NH4}^{Syn}$ was higher, an alternative state of coexistence could persist (Fig. 4B, zone i). However, this scenario would require *Synechococcus* to prefer $NH_4^+$ at low concentrations, which would be in violation of our incubation results.

Preferential grazing by zooplankton (e.g., nanoflagellates [29]) or virus (30, 31) may further ensure the coexistence of these species (32–35). However, our study shows that, without reliance on zooplankton behavior, *Prochlorococcus* and *Synechococcus* have inherent nutrient-based mechanisms for coexistence.

The lower *K* value for $NO_3^-$ in *Synechococcus* than that for *Prochlorococcus* predicts that their relative abundance will also depend strongly on the $NO_3^-/NH_4^+$ ratio of the nutrient supply (Fig. 5A). We tested this model prediction using field observations from the Hawaii Ocean Timeseries (HOT [36]) in the subtropical North Pacific (Fig. 5B). We find that the ratio cell counts of *Prochlorococcus* to *Synechococcus* exhibit a strong relationship to ambient $NO_3^-$ concentration (Fig. 5B). This finding is consistent with a previously observed positive relationship between nanomolar $NO_3^-$ concentrations at the surface and *Synechococcus* abundance but not for *Prochlorococcus* abundance (37–39). A *Synechococcus* bloom was also observed after the transient nanomolar increase of surface $NO_3^-$ in the stratified

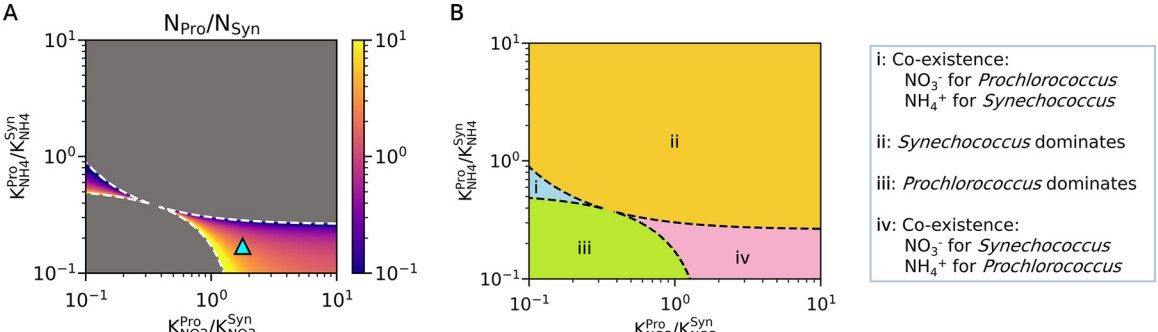

**FIG 4** Steady-state solution of the model for the coexistence of *Prochlorococcus* and *Synechococcus* based on the ratios of half-saturation constants for nitrogen sources. (A) Model results of the ratio of *Prochlorococcus* and *Synechococcus* expressed in terms of the proportion of total cellular nitrogen concentration in the water ($N_{Pro}/N_{Syn}$, see the color bar for values). The cyan triangle in A represents the optimized ratios of the half-saturation constants predicted from Fig. 2. The model output is based on the resource ratio of $NO_3^-$ and $NH_4^+$ that is 1:3 (for other ratios, see Fig. 5 and Fig. S6), which allows the coexistence of these organisms with the optimized ratios. Dashed curves indicate borders between coexistence and noncoexistence. (B) Explanation of each domain; coexistence occurs only in domains i and iv.

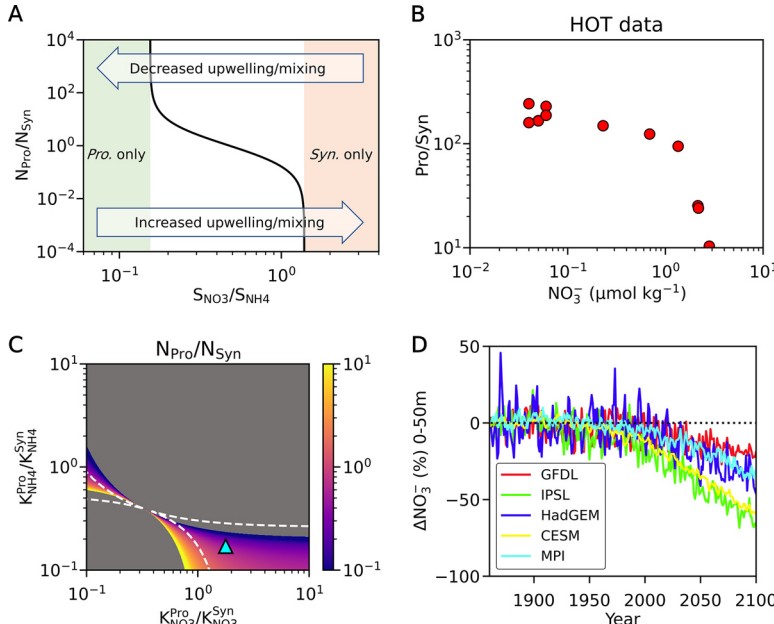

**FIG 5** Resource-ratio-based shifts of coexistence. (A) The shift in the ratio of *Prochlorococcus* and *Synechococcus* based on the resource ratios of $NO_3^-$:$NH_4^+$ ($S_{NO3}/S_{NH4}$) with the predicted half-saturation constants. (B) Relationship between the $NO_3^-$ concentrations and cell count ratios of *Prochlorococcus* and *Synechococcus* in HOT. (C) The effect of doubled $NO_3^-$ recourse. The plot shows the model results of the *Prochlorococcus* and *Synechococcus* ratio in terms of cellular N ($N_{Pro}/N_{Syn}$, see the color bar for values). Here, the resource ratio of $NO_3^-$:$NH_4^+$ is 2:3. Dashed curves indicate borders within which the coexistence occurs when the resource ratio of $NO_3^-$:$NH_4^+$ is 1:3 (same dashed curves as in Fig. 4A). The cyan triangle represents the ratios of predicted half-saturation constants for Fig. 2. (D) $NO_3^-$ concentration shifts predicted by various climate models (see Materials and Methods). The y-axis represents the fractional (%) change in surface nutrient concentration throughout the low latitudes (30°S-30°N, 0-50m).

Sargasso Sea (37). The stimulation of *Synechococcus* photosynthetic and growth activity after 20 nM $NO_3^-$ enrichment was later confirmed by an *in situ* bioassay experiment (38). While additional $NH_4^+$ input could have been added by the heterotrophic community or diazotrophs, the effect of extra $NH_4^+$ must have been small compared with 100 nM enrichment during our experiment and is unlikely to significantly affect our hypothesis (Fig. 1; Fig. S2). Taxon-specific differences in phytoplankton mortality could also be present, but their potential role in coexistence cannot be constrained by available data.

Shifts in the resource ratio $NO_3^-$:$NH_4^+$ can also alter the range of species traits that permit coexistence (Fig. 5C). When the $NO_3^-$ resource increases, the range of coexistence (iv in Fig. 4B; hereafter called domain iv) shifts to the left, favoring lower $K_{NO3}^{Pro}/K_{NO3}^{Syn}$ and $K_{NH4}^{Pro}/K_{NH4}^{Syn}$ for coexistence (Fig. 5C). On the other hand, when the $NH_4^+$ resource increases, the domain iv shifts to the right, favoring higher $K_{NO3}^{Pro}/K_{NO3}^{Syn}$ and $K_{NH4}^{Pro}/K_{NH4}^{Syn}$ for coexistence (Fig. S6A). Due to these shifts, at an increased $NO_3^-$ resource, we predict an increased ratio of *Synechococcus* at the predicted $K$ values (represented by cyan triangles in Fig. 4A and Fig. 5C), and a further increase in resource ratios of $NO_3^-$:$NH_4^+$ eventually leads to a predominance of *Synechococcus* (Fig. 4B, 5A; Fig. S6B).

The niche partitioning of Pro/Syn based on differential preferences for reduced and oxidized forms of N has important ecological and climatic implications. The $NO_3^-$/$NH_4^+$ ratio of nutrient supply can be influenced strongly by climate change, through its impact on ocean stratification, which reduces the supply of $NO_3^-$ to the photic zone of the low latitudes (40). This trend is supported by various climate modeling showing decreased $NO_3^-$ concentration in low latitude euphotic zones (Fig. 5D). According to our ecological modeling, such climate changes are likely to shift the community composition of these major biomes toward a greater preponderance of *Prochlorococcus* than that of *Synechococcus* (Fig. 5A). While long-term trends are not available to test this prediction, it is consistent with perturbations observed on shorter time scales in response to transient increases of surface $NO_3^-$ (2, 41).

However, how strongly climate change may alter the supply ratio of $NO_3^-$ to $NH_4^+$ contains substantial uncertainties, partially due to the difficulty in simulating $NH_4^+$ in the ocean (42). The preference of $NO_3^-$ by *Synechococcus* cannot be explained by the snapshot of vertical distribution in oligotrophic waters (39) being a marked *Prochlorococcus* abundance peak just above the nitracline, while for *Synechococcus*, the abundance peak is low all along the vertical profile (43, 44). This finding suggests we consider the nutrient history of phytoplankton, namely, frequency, scale, and duration of nutrient input (45). In fact, monthly observations at the Bermuda Atlantic Time-series Study (BATS) (2), as well as in the Gulf of Aqaba, Red Sea (41), observed spring blooms of *Synechococcus* when the water column was deeply mixed and the $NO_3^-$ supply from deep was active and observed *Prochlorococcus* domination during summer stratification, consistent with our model predictions (Fig. 5A). Increased primary production supported by *Synechococcus* in upwelling cyclonic eddies was reported from BATS (46) and off California (47), further supporting model simulations (Fig. 5A).

The predicted and observed preference of $NO_3^-$ by *Synechococcus* is qualitatively consistent with phylogenomic analysis, where *Synechococcus* possesses genes encoding both nitrate reductase (*narB*) and nitrite reductase (*nirA*), whereas these genes are patchy among *Prochlorococcus* strains (Fig. 6) (10, 19, 48–50). In addition to inorganic N, both *Prochlorococcus* and *Synechococcus* are capable of the uptake of organic forms of nitrogen, such as urea, similar to or better than $NH_4^+$ or $NO_3^-$ (19, 51), as well as amino acids (52–56) using high-affinity amino acid transporters at low ambient N concentrations (57–59). Culture studies of *Synechococcus* showed light-stimulated incorporation of amino acids (60). In addition, the presence of genes encoding transporters of organic compounds varies among strains of marine picocyanobacteria (55). In general, the amino acid incorporation was similar to or greater in *Prochlorococcus* than that of *Synechococcus* (55, 56). Further studies to estimate the effect of organic nitrogen on their growth will be important.

Changes in the $NO_3^-/NH_4^+$ supply ratio can be altered by ecological changes in addition to climatic changes. The low N environments dominated by these taxa are also prime habitats for $N_2$-fixing diazotrophs, which inject newly fixed $NH_4^+$ into surface waters. The presence of $N_2$-fixing organisms that provide $NH_4^+$ may favor *Prochlorococcus*, due to its lower $K$ value for $NH_4^+$, and as a result, *Synechococcus* may be outcompeted. This dynamic may underlie the correlations between *Prochlorococcus* and the filamentous diazotroph *Trichodesmium* reported from the oligotrophic South Pacific Ocean (61). Similarly, a positive relationship between *Prochlorococcus* and the unicellular diazotroph *Crocosphaera* has been suggested by showing analogous nutrient limitation in both *Crocosphaera* and *Prochlorococcus* (62), promoting the coexistence of these organisms. Distributions of both *Prochlorococcus* and *Crocosphaera* also show a positive correlation with temperature in oligotrophic oceans (3, 63). On the other hand, a negative relationship between the abundance of *Synechococcus* and nano-sized cyanobacteria (most likely *Crocosphaera*) (64) has been reported in the Pacific Ocean. Rising surface temperatures, rising atmospheric $CO_2$ concentrations, and strengthened water-column stratification (65) may favor diazotrophs (63, 66–68) in the future, supporting the predictions of *Prochlorococcus* domination (3) (Fig. 5A).

Overall, our observation-model synthesis suggests that the coexistence of the major marine phytoplankton is sustained by the differentiated nitrogen uptake. We were able to obtain this implication based on the high-precision measurement of nutrient concentration at nanomolar levels. Given the nutrient-depleted nature of the oligotrophic ocean and the shift in the nitrogen preference of *Synechococcus* from $NO_3^-$ to $NH_4^+$ at the nanomolar level, we propose that the high-precision measurements and studies focused on the nanomolar level concentrations are essential for elucidating the ecological dynamics and biogeochemical process in the oligotrophic ocean and predicting their future shift.

**Different uptake affinity to N in the content of picocyanobacterial evolution.** Marine *Synechococcus* and *Prochlorococcus* are estimated to have sequentially diverged

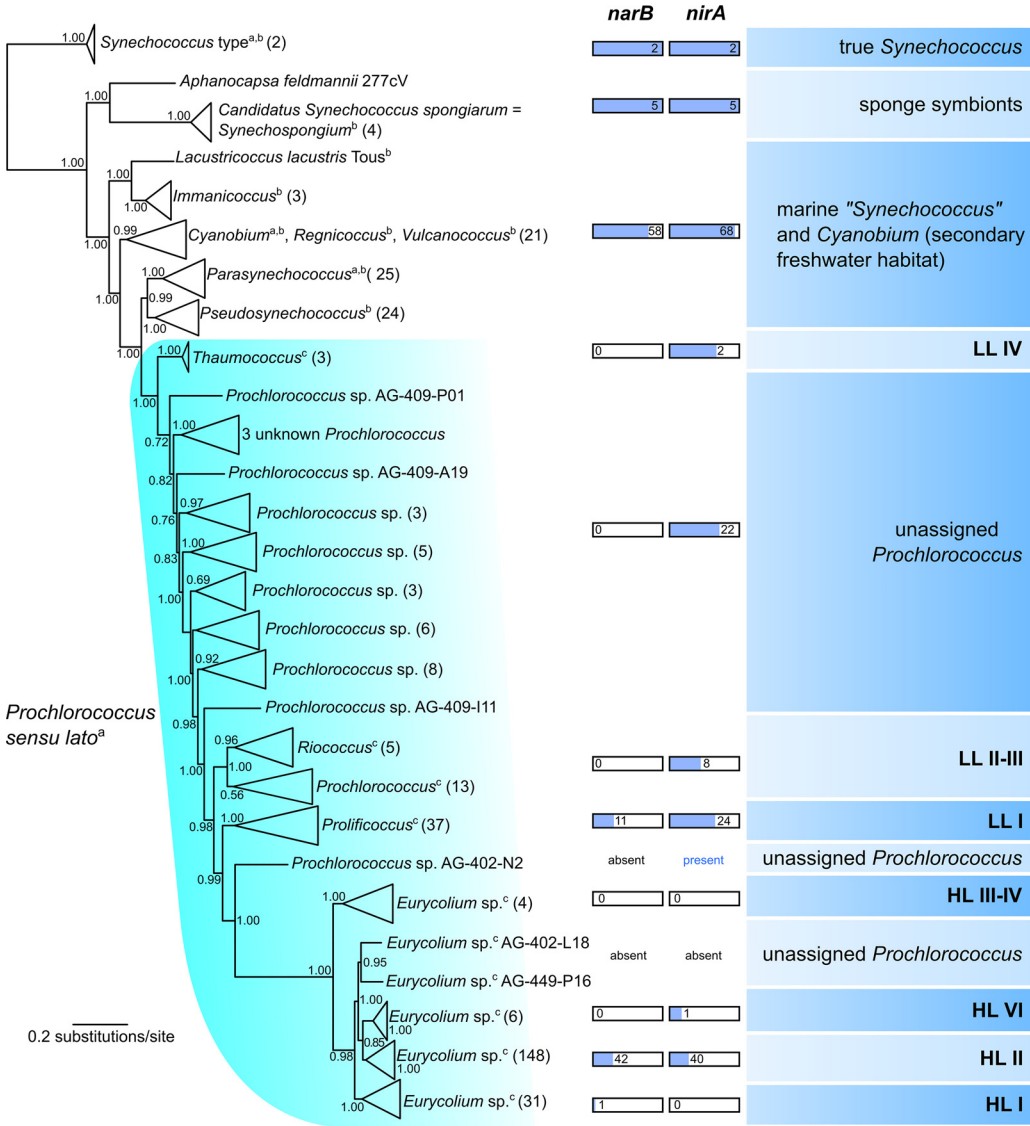

**FIG 6** Phylogenomic tree of marine picocyanobacteria. The tree is rooted by the freshwater core cluster of *Synechococcus* (type strain *S. elongatus* PCC 6301). Individual clades are annotated following the most recent taxonomic revisions. The a, b, and c in superscript in the figure represent references 70–72, respectively. Phylogenetic lineages are further classified using ecological groups of *Synechococcus* and *Prochlorococcus* (right); LL and HL stand for low-light- and high-light-adapted subclades of *Prochlorococcus* (18). The multigenus clade traditionally considered *Prochlorococcus sensu lato* is highlighted in cyan. The presence of genes encoding nitrate reductase (*narB*) and nitrite reductase (*nirA*) was assessed for the individual ecological groups and expressed using the total number of occurrences within the group along with a relative abundance scale bar. The tree is based on 120 conserved bacterial proteins using the approximately maximum likelihood method following the GTDB-Tk *de novo* pipeline (92). Bootstrap values are shown near the nodes, and numbers in brackets indicate the number of genomes within collapsed clades.

from a common ancestor during the Devonian, at about 413 million years ago (Mya), and Carboniferous, at around 360 Mya and 341 Mya, respectively (69). During these periods, a number of genus-level lineages likely evolved within both *Synechococcus* (e.g., *Parasynechococcus*, "*Pseudosynechococcus*") and *Prochlorococcus* (e.g., "*Thaumococcus*," "*Prolificoccus*," "*Riococcus*," and "*Eurycolium*") (70–72). Our reconstruction suggests that many other putative genera are still waiting to be recognized (Fig. 6). These lineages are in many aspects cryptic due to morphological simplification and genome streamlining, which are driven by convergent selective pressures, especially the nutrient deficiency addressed in this study (49). Nevertheless, individual ecotypes and even individual cells of picocyanobacteria harbor distinct sets of metabolic genes reflecting their ecological niche partitioning (73–75).

The ability to assimilate $NO_3^-$ is observed to vary widely among picocyanobacteria. Whereas most genomes of marine *Synechococcus* harbor both nitrate reductase and nitrite reductase genes, these genes exhibit a patchy distribution throughout both low-light and high-light adapted *Prochlorococcus* lineages (18, 50) (Fig. 6). According to previous studies, gene loss, as well as homologous recombination among closely related cell populations, is frequent in the genomic regions responsible for $NO_3^-$ acquisition in *Prochlorococcus* (50). Nitrate assimilation is energetically demanding and thus more likely to occur under high-light conditions near the water surface, but the concentration of $NO_3^-$ tends to increase with depth in oligotrophic oceans. This trade-off likely shapes the composition of *Prochlorococcus* populations by creating an equilibrium of $NO_3^-$ assimilating and nonassimilating subpopulations that coexist at a given depth and light intensity to maximize the effectivity of N uptake (50).

Phylogenomic analysis (Fig. 6) implies that mixed populations of $NO_3^-$ assimilating or nonassimilating "*Prochlorococcus*" cells corresponding to several different genus-level lineages may further influence the relative affinity of *Prochlorococcus* to different N substrates. This finding could be another mechanism underlying the coexistence of *Prochlorococcus* and *Synechococcus*, and their various phylogenetic sublineages, at different concentrations of available N forms. The relative contribution of genomic population diversity versus physiological affinity to specific forms of N to the coexistence of both picocyanobacteria was not addressed in the current study and thus warrants further investigation. With regard to the differential phylogeographic distribution of individual picocyanobacterial genotypes and ecotypes (76, 77), care must be taken when extrapolating the findings to oceanic regions outside Northwest Pacific, in which the verification of our hypothesis is still required.

**Conclusions.** Our study provides a quantitative explanation for the well-known coexistence of *Prochlorococcus* and *Synechococcus* and a new experimentally based test of a central prediction in the long-standing theory of resource competition. Nanomolar-scale nutrient addition captures the subtle but clear difference in nitrogen usage by these two organisms. We have used the data to parameterize a simple ecological model, showing that the specialization of nitrogen use ($NO_3^-$ versus $NH_4^+$) leads to the coexistence of these organisms. The model also predicts shifts in the balance of competing species across a range of resource supply ratios, yielding the exclusion of *Prochlorococcus* or *Synechococcus* at high and low ratios of $NO_3^-$: $NH_4^+$, respectively. These model predictions resemble measured variations in species abundances and their covariation with $NO_3^-$ and are also qualitatively consistent with previous observations of physical phenomena that naturally perturb the supply ratio of reduced to oxidized N. Evaluating the predicted shifts in the niche of these dominant phytoplankton will require the incorporation of differential nutrient specialization and nutrient resource fluxes in global biogeochemistry and climate models.

## MATERIALS AND METHODS

**Observations and experiments.** A series of bioassay experiments was carried out using natural phytoplankton assemblages collected at a station in the subtropical Northwestern Pacific (12°N, 135°E) from 6 to 25 June 2008 during the MR08-02 cruise on the R/V *MIRAI*. Detailed experimental procedures are described in reference 16. Five macronutrient (N and P)-addition bioassays (M1 to M5) and three Fe-addition bioassays (Fe1 to Fe3) were conducted to elucidate the availability and preference of natural phytoplankton communities for N, P, and Fe sources (Table S7 and S8 in reference 16).

**Seawater collection.** Water samples were collected from a 10-m depth at 12:30 h local time using Teflon diaphragm pump system consisting of Teflon tubing and associated plastic ware. All components of this pump system and associated plastic were washed overnight in a neutral detergent, washed with HCl and $HNO_3$, rinsed with heated Milli-Q water, and flushed with seawater for 30 min immediately prior to sample collection. For the nutrient-addition bioassays, 4-L polycarbonate incubation bottles and other plastic instruments were rinsed overnight in a neutral detergent, followed by 0.3 N HCl, and rinsed with Milli-Q water. For Fe-addition experiments, the 2-L polycarbonate incubation bottles had been cleaned according to reference 78. Other polyethylene and Teflon lab wares were cleaned according to reference 79. All washing procedures were carried out in an onshore class-1000 clean air room, and plastic gloves were worn during these operations.

To reduce grazing pressure, we prefiltrated seawater samples prior to the bioassay setup. For the bioassay treatments with N and P addition, water was filtered through an acid-cleaned 1-$\mu$m in-line

cartridge filter (Micropore EU, ORGANO). For Fe additions, seawater was prefiltered through a 10-$\mu$m filter of the same manufacturer. The prefiltered water was then dispensed into the corresponding bioassay incubation bottles.

Iron concentrations of the seawater were measured as total iron (TFe), on the whole water samples collected directly from the pump system, and as dissolved iron (DFe), on the 125 mL of seawater collected in low-density polyethylene bottles (Nalgen; Nalge Nunc International) and filtered through an acid-cleaned 0.22-$\mu$m pore filter (Millipack-100; Millipore). All TFe and DFe samples were acidified with HCl to pH <1.5 and stored at room temperature for at least 1 year.

Triplicate samples for the $NO_3^-$ + $NO_2^-$ (N+N), $NH_4^+$, urea and soluble reactive phosphorus (SRP) (12, 80) analysis were collected in 100 mL of 0.1 N HCl-rinsed polyethylene bottles. All samples were analyzed onboard, with the exception of urea, which was measured only in the urea treatment. Upon collection, all samples were stored at −20°C until analysis.

**Nutrient addition bioassay set up and incubation.** A total of eight bioassays were carried out, of which five were macronutrient bioassays with additions of different forms of nitrogen or phosphate (M1 to M5) and three were Fe-addition bioassays (Fe1 to Fe3). For macronutrient bioassays, prefiltered seawater was dispensed into 4-L polycarbonate bottles. Five different triplicate treatments were set up, as follows: one control without nutrient addition; three treatments with 100 nM addition of N, whether as $NaNO_3$, $NH_4Cl$, or urea; and one treatment with 10 nM $NaH_2PO_4$ (P). Bottles were incubated on deck in flow-through seawater tanks covered with a neutral density screen to attenuate light intensity to 50% of its corresponding surface value. Macronutrient bioassays lasted 3 days with daily sample collections to follow changes in phytoplankton pigments by high-pressure liquid chromatography (HPLC) and community composition by flow cytometry (FCM). For the Fe-addition bioassays, the filtrate was poured into 2-L polycarbonate bottles that had been cleaned according to methods of reference 78. Five duplicate treatments were set up, as follows: controls without any nutrient addition, phosphate additions with 10 nM $NaH_2PO_4$, iron addition with 1 nM $FeCl_3$, an Fe+P treatment with 1 nM $FeCl_3$ and 10 nM $NaH_2PO_4$, and Fe+N treatment with amendment of 1 nM $FeCl_3$ and 100 nM $NaNO_3$. To all treatments containing iron addition, EDTA (1 nM) was added as a buffer. Fe-addition treatments were done in an onboard class-100 clean air room. Bottles for the iron-addition bioassays were also incubated in on-deck flow-through seawater tanks covered with a neutral density screen to attenuate light intensity to 50% of its corresponding surface value. Iron-addition bioassays lasted for 5 days, monitoring TFe, DFe, and phytoplankton community composition on days 0, 1, 3, and 5.

**Macronutrient analysis.** Concentrations of $NO_3^-$ + $NO_2^-$ (N+N), $NH_4^+$, SRP, and urea were measured using a high-sensitivity colorimetric approach with an autoanalyzer II instrument (Technicon) and liquid waveguide capillary cells (World Precision Instruments, USA) as outlined (13). Urea concentrations were analyzed by the diacetyl monoxime method (81). Detection limits of $NO_3^-$ + $NO_2^-$, $NH_4^+$, and SRP were 3, 6, and 3 nM, respectively.

**TFe and DFe concentrations.** Dissolved Fe(III) in seawater samples was determined using catalytic cathodic stripping voltammetry with a detection limit of 6 pM using the approach of reference 82. No contamination during sampling and incubation was detected.

**Flow cytometry (FCM).** *Prochlorococcus* and *Synechococcus* were identified using flow cytometry (FCM) based on cell size and chlorophyll or phycoerythrin fluorescence. Aliquots of 4.5 mL were preserved in glutaraldehyde (1% final concentration), flash frozen in liquid $N_2$, and stored at −80°C until analysis by flow cytometer (PAS-III; Partec, GmbH, Münster, Germany) equipped with a 488-nm argon-ion excitation laser (100 mW) on land. Forward- and side-angle scatter (FSC and SSC, respectively), red fluorescence (>630 nm, FL3), and orange fluorescence (570 to 610 nm, FL2) were recorded. *Synechococcus* and *Prochlorococcus* were distinguished using FloMax (Partec, GmbH) based on their autofluorescence properties and their size.

**Statistical analysis.** Phytoplankton cell densities of each bioassay were first compared between treatments using repeated measures analysis of variance (RM-ANOVA) with nutrient treatments as the between-subjects factor (5 levels) and time (4 levels) as the within-subjects factor. Treatment effects were considered significant if the $P$ value was <0.05. Then, means between five treatments were compared by a *post hoc* Turkey test ($n$ = 3 replicates per treatment throughout, degrees of freedom = 40).

**The ecological model of *Prochlorococcus* and *Synechococcus*.** The model follows simple balances of cell densities and nutrient concentrations as used in resource competition theory (6, 7, 21, 22). This theory is suitable for predicting the ecological niches of phytoplankton in aquatic systems, such as diatoms (7) and $N_2$ fixers (6, 21, 22), based on different nutrient uptake behavior. We used the following four key equations:

$$\frac{dX_{Pro}}{dt} = \left( V_{maxNO3}^{Pro} \frac{[NO_3^-]}{[NO_3^-] + K_{NO3}^{Pro}} + V_{maxNH4}^{Pro} \frac{[NH_4^+]}{[NH_4^+] + K_{NH4}^{Pro}} - m_{Pro} \right) X_{Pro} \tag{1}$$

$$\frac{dX_{Syn}}{dt} = \left( V_{maxNO3}^{Syn} \frac{[NO_3^-]}{[NO_3^-] + K_{NO3}^{Syn}} + V_{maxNH4}^{Syn} \frac{[NH_4^+]}{[NH_4^+] + K_{NH4}^{Syn}} - m_{Syn} \right) X_{Syn} \tag{2}$$

$$\frac{d[NO_3^-]}{dt} = -V_{maxNO3}^{Pro} \frac{[NO_3^-]}{[NO_3^-] + K_{NO3}^{Pro}} Q_N^{Pro} X_{Pro} - V_{maxNO3}^{Syn} \frac{[NO_3^-]}{[NO_3^-] + K_{NO3}^{Syn}} Q_N^{Syn} X_{Syn} + S_{NO3} \tag{3}$$

$$\frac{d[NH_4^+]}{dt} = -V_{maxNH4}^{Pro}\frac{[NH_4^+]}{[NH_4^+]+K_{NH4}^{Pro}}Q_N^{Pro}X_{Pro} - V_{maxNH4}^{Syn}\frac{[NH_4^+]}{[NH_4^+]+K_{NH4}^{Syn}}Q_N^{Syn}X_{Syn} + S_{NH4} \tag{4}$$

where $X_i$ (cell L$^{-1}$) = abundance of phytoplankton $i$ ($i = Pro, Syn$; *Prochlorococcus* and *Synechococcus*, respectively)

$t$ (d) = time

$V_{maxj}^i$ (d$^{-1}$) = maximum uptake rate of phytoplankton $i$ for nutrient $j$ ($j = NO_3^-, NH_4^+$)

$[j]$ (nmol L$^{-1}$) = nutrient concentration

$K_j^i$ (nmol L$^{-1}$) = half-saturation constant of nutrient $j$ for phytoplankton $i$

$m_i$ (d$^{-1}$) = mortality rate of phytoplankton $i$

$Q_N^i$ (nmol cell$^{-1}$) = cellular nitrogen content of phytoplankton $i$

$S_j$ (nmol L$^{-1}$ d$^{-1}$) = resource term for nutrient $j$

The cell densities are the outcome of the balances between cellular growth and mortality (equation 1 and 2). The balances of $NO_3^-$ and $NH_4^+$ are represented by the uptake by cells and a constant source of each nutrient (equation 3 and 4). For $Q_N^i$, we used the average values from experiments M1 to M3, which were estimated based on the FCM-analyzed cell size with the reference carbon per cell volume (83) and carbon-to-nitrogen ratios (84). We first manually parameterized $V_{maxj}^i$, $K_j^i$, $m_i$, and $S_j$ as well as initial values for $X_i$ and $[j]$ (hereafter, predicted parameters) in order for $X_i$ and $[j]$ to closely represent observations (Fig. 2; Fig. S3). After that step, we applied the Metropolis-Hastings algorithm (85, 86) as used in reference 87 and 88 to improve the parameterization of $V_{maxj}^i$, $K_j^i$, and $m_i$. Parameter values and initial values are in Table S4 and S6, respectively. We limited the application of the algorithm to these parameters because the number of parameters is already large. After the parameters were selected, we solved these equations under the steady state to obtain $X_i$ values using Mathematica 11.3 (89) and plotted $Q_N^{Pro}X_{Pro}/Q_N^{Syn}X_{Syn}$ for different resource ratios (Fig. 4A, 5A, 5C; Fig. S6). Because the equations are based on the existence of these two organisms, valid solutions are obtained only where these organisms coexist. For Fig. 4A and 5C and Fig. S6A and B, we varied the ratios of half-saturation constants, using $NO_3^-$: $NH_4^+$ resource ratios of 1:3, 2:3, 1:6, and 3:1, respectively.

We chose these equation formulas given the currently available data. The addition of $NH_4^+$ and $NO_3^-$ may not strictly represent the physiology these organisms. For example, there are finer-scale intracellular molecular interactions than those represented here. For example, $NO_3^-$ may become a source for $NH_4^+$, and maximum growth rate may be constrained separately. Despite the simplification, however the model reproduced the population data (Fig. 2) under N limitation, which accounts for a large part of the *Prochlorococcus* and *Synechococcus* habitats (3, 90). Thus, we hypothesized that more detailed intracellular mechanisms may have only a secondary effect. Resolving intracellular mechanisms would require a number of additional parameters, which may not be as well constrained as those we use here; further experiments would be required for more detailed physiological analyses.

**Nutrient trend analysis.** We analyzed surface nutrient trends from projections of future climate states under the Representative Concentration Pathway (RCP) 8.5 greenhouse gas emissions scenarios from 5 different Earth System Models (ESMs) participating in the Climate Model Intercomparison Project (CMIP5) (91) (see Table S7 in the supplemental material). The past and future climate trajectories in ESMs are forced by historical greenhouse gas emissions until the early 21st century and then by the RCP8.5 emissions scenario, which reaches a nominal radiative forcing of 8.5 W/m$^2$ in 2100 (Table S7).

**Phylogenetic analysis.** To assess the evolutionary history of picocyanobacteria, we constructed a phylogenetic tree utilizing a *de novo* workflow available in the Genome Taxonomy Database (GTDB) toolkit (92, 93). GTDB-Tk v1.3.0, employing the R05-RS95 release of GTDB from July 2020, was used to produce a standardized approximately maximum likelihood phylogenomic tree (FastTree2) (94) based on 120 concatenated conserved bacterial markers. The tree spanned all known genome-sequenced picocyanobacterial clades and was rooted by a closely related outgroup taxon, *Synechococcus elongatus*. To assess the potential of picocyanobacteria to assimilate $NO_3^-$ (and nitrite: $NO_2^-$), all genomes included in the analysis were screened for the presence of the nitrate reductase (*narB*) and ferredoxin-nitrite reductase (*nirA*) genes. The search for *narB* and *nirA* homologues was performed using BLASTp and tBLASTn with query proteins mined from *S. elongatus* PCC 6301 (genome accession AP008231.1) and default parameters; the resulting hits were verified using the conserved domain search available in NCBI (95). Taxonomic grouping of picocyanobacteria was annotated following recent studies (70–72), and individual phylogenetic lineages were subsequently assigned to the traditional ecological scheme of low-light-versus high-light-adapted ecotypes (10, 50). Table S8 shows a list of strains and genome assemblies used for phylogenomic reconstruction in Fig. 6, and their placement in clades as dipicted therein.

**Code availability.** The model has been written in Python 3 and is freely available in Zenodo online at https://zenodo.org/record/4568561.

**Data availability.** The data used to generate the graphs presented in the main figures can be found in Data Set S1 in the supplemental material or online at Zenodo (https://zenodo.org/record/4568561). All other data that support the findings of this study are available on request from the corresponding author.

## SUPPLEMENTAL MATERIAL

Supplemental material is available online only.

**SUPPLEMENTAL FILE 1**, XLSX file, 0.01 MB.

**SUPPLEMENTAL FILE 2**, PDF file, 1.9 MB.

## ACKNOWLEDGMENTS

We thank the captain, crew, and technicians of the R/V *MIRAI* for assistance and support during the research cruise. We acknowledge the World Climate Research Programme's Working Group on Coupled Modeling, which is responsible for the CMIP, and we thank the climate modeling groups (listed in Table S8) for producing and making available their model output.

This research was financially supported by MEXT grants for Scientific Research on Innovative Areas (24121001 and 24121005 to K.F.), Czech Research Foundation GAČR (project 20-17627S to O.P. and T. Masuda), JSPS KAKENHI (project 20H03059 and 22H05201 to T. Masuda), the Simons Foundation (Life Sciences-Simons Postdoctoral Fellowships in Marine Microbial Ecology, Award 544338, to K.I.), and the Gordon and Betty Moore Foundation (grant no. 3775 to C.D.). This publication uses data from Hawaii Ocean Time-series observations supported by the U.S. National Science Foundation under award no. 1756517.

T. Masuda, K.F., and S.T. designed the *in situ* microcosm experiments. T. Masuda, T.K., T.S., and T. Matsui carried out the experiment and analyzed data supervised by K.F., S.T., and K.S. T. Masuda and K.I. shaped the concept of the study with the supervision of O.P. and C.D. K.I. developed the model with the supervision of C.D. C.D. analyzed the data from HOT and CMIP models. J.M. conducted phylogenetic analysis. T. Masuda, K.I., and J.M. wrote the original draft with substantial input from all the authors.

We declare no competing financial interests.

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
