## [Reviewer comments · Microbiology Spectrum]

Microbiology Spectrum

Coexistence of dominant marine phytoplankton sustained by nutrient specialization

Takako Masuda, Keisuke Inomura, Jan Mares, Taketoshi Kodama, Takuhei Shiozaki, Takato Matsui, Koji Suzuki, Shigenobu Takeda, Curtis Deutsch, Ondřej Prášil, and Ken Furuya

Corresponding Author(s): Takako Masuda, Kokuritsu Kenkyu Kaihatsu Hojin Suisan Kenkyu Kyoiku Kiko

Review Timeline:

Submission Date:	October 1, 2022
Editorial Decision:	November 9, 2022
Revision Received:	May 16, 2023
Accepted:	June 7, 2023

Editor: Adriana Lopes dos Santos

Reviewer(s): Disclosure of reviewer identity is with reference to reviewer comments included in decision letter(s). The following individuals involved in review of your submission have agreed to reveal their identity: Denise Rui Ying Ong (Reviewer #1)

Transaction Report:

DOI: <https://doi.org/10.1128/spectrum.04000-22>

November 9, 2022

Dr. Takako Masuda
Japan Fisheries Research and Education Agency
Shiogama
Japan

Re: Spectrum04000-22 (Coexistence of dominant marine phytoplankton sustained by nutrient specialization)

Dear Dr. Takako Masuda:

Please see the comments of reviewer below. As editor, I would like to see the diversity within *Prochlorococcus* and *Synechococcus* being explored and discussed. This impacts the conclusion of the study, as the results observed in the Northwestern Pacific cannot be generalized to other environmental context.

The main caveat is that the diversity within *Prochlorococcus* and *Synechococcus* is not discussed. In my opinion, this impacts the conclusion of the study, as the results observed in the Northwestern Pacific cannot be generalized to other environmental context. This diversity aspect is not even mentioned in the manuscript.

Thank you for submitting your manuscript to *Microbiology Spectrum*. When submitting the revised version of your paper, please provide (1) point-by-point responses to the issues raised by the reviewers as file type "Response to Reviewers," not in your cover letter, and (2) a PDF file that indicates the changes from the original submission (by highlighting or underlining the changes) as file type "Marked Up Manuscript - For Review Only". Please use this link to submit your revised manuscript - we strongly recommend that you submit your paper within the next 60 days or reach out to me. Detailed instructions on submitting your revised paper are below.

Link Not Available

Below you will find instructions from the *Microbiology Spectrum* editorial office and comments generated during the review.

Sincerely,

Adriana Lopes dos Santos

Editor, *Microbiology Spectrum*

Journals Department
Reviewer comments:

Reviewer #1 (Comments for the Author):

Overall, I found the paper to be well written. It was clearly structured and therefore easy to follow through. The objectives of the study were clearly listed in the introduction and answered in the results and discussion. The data presentation and use of statistics was suitable and useful to understand the data collected. I also thought that the discussion was well supported with previous studies. I especially found the use of phylogenetic analysis of genes involved in nitrate utilization to be useful supporting information.

I have two major comments.

- Would there be any niche partitioning by depth between *Prochlorococcus* and *Synechococcus*, as the concentration of nitrate is lower at the surface and increases with depth? Are there any comparisons between cell concentration of *Prochlorococcus* and *Synechococcus* at the deep chlorophyll maximum and correlating to the relative concentrations of nitrate to ammonia?
- Please provide some references for line 197 stating that ocean stratification reduces supply of nitrate to photic zone of low latitudes. What is the predicted magnitude of decrease nitrate and how does it relate to the concentrations of nitrate and ammonia used in this study? Since *Synechococcus* can utilize nitrate, as proven from culture based studies, the magnitude of change might not be as large as suggested from the data.

I also have a few minor comments.

- Fig 1 - what does the dotted line represent?
- Fig 2 - are the bars around the data points representing standard error?
- Fig 3 and 4 - resolution of the figures are too low to read the axis.
- The manuscript has a few informal language use eg. "can't". Please change to formal language.
- Line 125 - is there another subheading that you could give this section for it to be more representative? One suggestion I can make is "Ecological model for N uptake preferences"
- Line 134 - There seems to be formatting error for the sentence "K for NO₃-Similarly..."
- I may have missed this in the text, but what is the pore size of the filter used to prefilter seawater samples to reduce grazing pressure?

Reviewer #2 (Comments for the Author):

In their article Coexistence of dominant marine phytoplankton sustained by nutrient specialization, Masuda and co-authors explore a potential hypothesis to explain the co-existence of the two genera of marine picocyanobacteria *Synechococcus* and *Prochlorococcus* in the the subtropical Northwestern Pacific. Through a set of on-deck incubation, they show that *Prochlorococcus* grows faster after NH₄⁺ addition than after NO₃⁻ addition, while the opposite pattern is observed for *Synechococcus*. Based on these incubation results, they then fit a simple growth model and estimate the nitrogen assimilation parameters associated to each genus. These parameters are used to discuss what allows for the coexistence of the two genera. The main result is that coexistence is permitted by different affinities of *Prochlorococcus* and *Synechococcus* to nitrate and nitrite.

While it is long known that *Prochlorococcus* and *Synechococcus* can co-occur, and that *Prochlorococcus* dominates in oligotrophic waters and *Synechococcus* in richer waters, to my knowledge no study has tried to find a mechanistic explanation for their coexistence. The two organisms and their physiologies are so different that it is probably a bit simplistic to reduce their coexistence to the use of different forms of nitrogen, but it is nonetheless interesting to highlight their difference in this regard. While the description of the results of on-deck incubations is relatively clear, the reasoning of the authors is sometimes hard to follow in the description of the model and its outputs, and the discussion fails to take into account the current knowledge of *Prochlorococcus* and *Synechococcus* diversity. These aspects are further developed in my comments below.

Main comments:

- Throughout the manuscript, the authors refer to *Synechococcus* and *Prochlorococcus* as different species, and explore the question of species coexistence. While the taxonomy of cyanobacteria still needs revision, neither *Synechococcus* nor *Prochlorococcus* can be considered as species-level taxonomic groups according to current genomic standard (see e.g. Salazar et al. 2020, Tschoeke et al. 2020, Doré et al. 2020), and they should be referred to as different genera. Both genera encompass a wide diversity described by an extensive literature, and the variability N acquisition abilities within each genus is well documented both by culture experiments (e.g. Moore et al 2002) and genomic analyses (e.g. Scanlan et al. 2009, Martiny et al 2009, Partensky and Garczarek 2010, Doré et al 2020). As a consequence, the results obtained with on-deck incubations are only valid for the community present at the sampling sites, and cannot be generalized to other environmental contexts. Different strains of *Synechococcus* and *Prochlorococcus* most likely have different half-saturation constants (K) for NO₃ and NH₄. The strain variability should thus at least be discussed somewhere in the manuscript.

- I don't understand the rationale behind the variation of K_{pro}/K_{syn} in Figures 3, 4 and S5. I understand that only some pairs of K_{pro} and K_{syn} for NO₃ and NH₄ allow for their coexistence, but these parameters have been determined using the model fit to the data, so why having them vary? What does that represent biologically?

I find more interesting to see in which NO₃:NH₄ ratios *Synechococcus* and *Prochlorococcus* can coexist, given the parameters estimated by the model, as shown in Fig 4A.

Line by line comments:

- L.39 "higher preference": could be more specific by saying "higher affinity" since the difference is in K. In addition, the authors did not test for the preference (which would mean to grow the bacteria with both nutrients, and see which one is used) but for the efficiency of growth with each nutrient. Due to transcriptomic regulation, if both were present at the same time, I am not sure that *Synechococcus* would prefer NO₃ over NH₄.
- L. 41-42: "is sufficient to explain their coexistence". I'm not sure that we can say it is sufficient. The model demonstrates that coexistence is possible when they have different nutrient affinities.
- L. 45 "species": as mentioned above, *Synechococcus* and *Prochlorococcus* are different genera. I would suggest to replace species by genera here and in the rest of the manuscript when referring to *Prochlorococcus* and *Synechococcus*.
- L. 57 "Nutrient utilization is a critical factor in microbial co-existence." It would be nice to have a citation from literature for this affirmation.
- L.64 Reference 7 (Martiny et al., 2009) concerns *Prochlorococcus* (and *Synechococcus*). This would be clearer if it was stated in the sentence, as the previous sentences are much more general.
- L. 83-84 The sentence reads a bit odd. Do you mean "to verify that only N was limiting"?
- L. 84 Fig S1 suggests that for *Synechococcus*, adding Fe + NO₃⁻ has more effect than adding NO₃⁻ only. Do you think there is a co-limitation?
- L. 86-87 "Throughout the observation": Table S1 in ref 14 only shows the initial nutrient concentrations, so I'm not sure what "Throughout the observation period" refers to. Or maybe the authors meant Fig S1 in ref 14? This can also be seen in Fig S2 of this manuscript.
- L. 93-94: the results in Table S2 show that in many replicates, urea is at least as good, and sometimes better than NH₄⁺ to stimulate *Prochlorococcus* growth (and similarly, sometimes as good or better than NO₃ for *Synechococcus*). I think this should be described and commented in the results, to explain why it was not included in the model (maybe because it was non-discriminant between the two organisms?).
- L. 108-123: The whole paragraph is hard to follow and needs rephrasing. Main ideas: it is energetically unexpected to prefer NO₃ over NH₄. This is not the first observation, but there is a lack of good explanation, both in *Synechococcus* and in other phytoplankton organisms.
- L. 132-133: it seems logical that when reversing the parameters that have been optimized for a given species, the fit is bad.
- L. 138-140: I am not sure to understand this sentence. Table S4 does not show other parameters, and the authors probably mean Table S3. But even in Table S3, only the mortality rates seem somewhat similar between the organisms (the quotas, v_{max} and K are all different)
- L. 140-142: I do not understand this sentence. I think the idea is that the differences in v_{max} are negligible compared to differences in K (in terms of their effect on the model output), which I agree with, but the sentence needs rephrasing. The following sentence (about affinity) expresses this idea well.
- L. 143 I would add the description of this measure of "affinity", as described in ref 31: "a better indicator of nutrient uptake efficiency than Ks"
- L. 147-148: out of curiosity, have some K values been measured from culture experiments in the literature for these organisms?
- L. 152 As mentioned above, I'm not sure to understand the rationale behind the variation of K ratios. In addition, why are the solutions where one organism outcompetes the other considered as invalid?
- L. 163 Differential mortality could also be due to virus predation.
- L. 179-181 I am not sure to understand why the use of amino-acids by *Prochlorococcus* is discussed at this point of the manuscript. But I agree that the use of other sources of N (amino-acid, cyanate, urea...) by both *Prochlorococcus* and *Synechococcus* should be discussed in more depth, especially since their genomic potential for N assimilation is well described.
- L. 195-197 This sentence needs a reference.
- L. 208-210 This difference (and the variability within *Prochlorococcus*) has been known for a long time. Please cite the literature accordingly (e.g Moore et al. 2002, Martiny et al. 2009, Scanlan 2009, Partensky and Garczarek, 2010)
- L. 323: why were only NO₃⁻ and NH₄⁺ chosen to be part of the model (and not urea, for which data was also available?)
- L. 346-347: this is a bit unclear, maybe change to "For Fig. 3A, 4C, S5A, S5B, we varied the ratios of half-saturation constants, using NO₃⁻:NH₄⁺ resource ratios of 1:3, 2:3, 1:6 and 3:1, respectively"
- L. 348-354 This reads more like a discussion, and should to my opinion be included in the main text.
- L. 396 "For N and P addition, data is based on Exp. M1-M3, where the initial concentrations of N were small": since the initial concentrations of N are used to select which data are represented on the figure, as well as for parametrizing the model (e.g. N quotas per cell), I believe that table S1 in ref 14 should be included in this manuscript as a supplemental table.
- L. 398 "Note that added NH₄⁺ was depleted on the third day": again, this is an important information to interpret the results, and I believe that Fig S1 in ref 14 should be included in this manuscript as well. Or otherwise refer to Fig S2.
- L.411-412 Please explain in the legend (and maybe in the main text as well) why a ratio of 1:3 was used as a reference.
- Fig S1 legend "*Prochlorococcus* prefers NH₄⁺ while, *Synechococcus* prefers NO₃⁻." From what I see, I would include urea in this sentence. In addition, I don't think that these incubations tested for preference, but rather that *Prochlorococcus* was more efficient in growing on NH₄⁺ and urea, and *Synechococcus* more efficient in growing on NO₃⁻ (and urea). Due to transcriptomic regulation, if both were present at the same time, I am not sure that *Synechococcus* would prefer NO₃ over NH₄.

Typos

- L. 130: Table S6 does not show K values. The authors probably meant Table S3.
- L. 133-134 "On the other hand, K for NO₃-": this is probably a typo to be removed.
- L. 141: it is table S3, not S5.
- L. 144: Table S4, not S5.
- L. 351 "and model the model": typo.
- L. 403 Points in figure 2 are huge (in Prochlorococcus, they are almost 10000 cells/mL wide). I would suggest to reduce point size.
- L. 411-412 "The model output is based on the resource ratio of NO₃- and NH₄+ is 1:3 (for other ratios, see Figure 3, S5)." There are 2 typos here (one extra "is", and Figure 3 is mentioned in its own legend!).
- Table S2: typo in M1, day3, Prochlorococcus: (Cont = PO₄) < (NO₃ = Urea) < Urea. Typo for Fe1, Fe2, Fe3 for Day1 in Prochlorococcus and Synechococcus: includes NO₃ and NH₄ and Urea instead of Fe and PO₄.

Staff Comments:

Preparing Revision Guidelines

Please return the manuscript within 60 days; if you cannot complete the modification within this time period, please contact me. If you do not wish to modify the manuscript and prefer to submit it to another journal, please notify me of your decision immediately so that the manuscript may be formally withdrawn from consideration by Microbiology Spectrum.

In their article *Coexistence of dominant marine phytoplankton sustained by nutrient specialization*, Masuda and co-authors explore a potential hypothesis to explain the co-existence of the two genera of marine picocyanobacteria *Synechococcus* and *Prochlorococcus* in the the subtropical Northwestern Pacific. Through a set of on-deck incubation, they show that *Prochlorococcus* grows faster after NH₄⁺ addition than after NO₃⁻ addition, while the opposite pattern is observed for *Synechococcus*. Based on these incubation results, they then fit a simple growth model and estimate the nitrogen assimilation parameters associated to each genus. These parameters are used to discuss what allows for the coexistence of the two genera. The main result is that coexistence is permitted by different affinities of *Prochlorococcus* and *Synechococcus* to nitrate and nitrite.

While it is long known that *Prochlorococcus* and *Synechococcus* can co-occur, and that *Prochlorococcus* dominates in oligotrophic waters and *Synechococcus* in richer waters, to my knowledge no study has tried to find a mechanistic explanation for their coexistence. The two organisms and their physiologies are so different that it is probably a bit simplistic to reduce their coexistence to the use of different forms of nitrogen, but it is nonetheless interesting to highlight their difference in this regard. While the description of the results of on-deck incubations is relatively clear, the reasoning of the authors is sometimes hard to follow in the description of the model and its outputs, and the discussion fails to take into account the current knowledge of *Prochlorococcus* and *Synechococcus* diversity. These aspects are further developed in my comments below.

Main comments:

- Throughout the manuscript, the authors refer to *Synechococcus* and *Prochlorococcus* as different species, and explore the question of species coexistence. While the taxonomy of cyanobacteria still needs revision, neither *Synechococcus* nor *Prochlorococcus* can be considered as species-level taxonomic groups according to current genomic standard (see e.g. Salazar *et al.* 2020, Tschoeke *et al.* 2020, Doré *et al.* 2020), and they should be referred to as different genera. Both genera encompass a wide diversity described by an extensive literature, and the variability N acquisition abilities within each genus is well documented both by culture experiments (e.g. Moore *et al.* 2002) and genomic analyses (e.g. Scanlan *et al.* 2009, Martiny *et al.* 2009, Partensky and Garczarek 2010, Doré *et al.* 2020). As a consequence, the results obtained with on-deck incubations are only valid for the community present at the sampling sites, and cannot be generalized to other environmental contexts. Different strains of *Synechococcus* and *Prochlorococcus* most likely have different half-saturation constants (K) for NO₃ and NH₄. The strain variability should thus at least be discussed somewhere in the manuscript.
- I don't understand the rationale behind the variation of K_{pro}/K_{syn} in Figures 3, 4 and S5. I understand that only some pairs of K_{pro} and K_{syn} for NO₃ and NH₄ allow for their coexistence, but these parameters have been determined using the model fit to the data, so why having them vary? What does that represent biologically?
I find more interesting to see in which NO₃:NH₄ ratios *Synechococcus* and *Prochlorococcus* can coexist, given the parameters estimated by the model, as shown in Fig 4A.

Line by line comments:

- L.39 “higher preference”: could be more specific by saying “higher affinity” since the difference is in K. In addition, the authors did not test for the preference (which would mean to grow the bacteria with both nutrients, and see which one is used) but for the efficiency of growth with each nutrient. Due to transcriptomic regulation, if both were present at the same time, I am not sure that *Synechococcus* would prefer NO₃ over NH₄.
- L. 41-42: “is sufficient to explain their coexistence”. I’m not sure that we can say it is sufficient. The model demonstrates that coexistence is possible when they have different nutrient affinities.
- L. 45 “species”: as mentioned above, *Synechococcus* and *Prochlorococcus* are different genera. I would suggest to replace species by genera here and in the rest of the manuscript when referring to *Prochlorococcus* and *Synechococcus*.
- L. 57 “Nutrient utilization is a critical factor in microbial co-existence.” It would be nice to have a citation from literature for this affirmation.
- L.64 Reference 7 (Martiny *et al.*, 2009) concerns *Prochlorococcus* (and *Synechococcus*). This would be clearer if it was stated in the sentence, as the previous sentences are much more general.
- L. 83-84 The sentence reads a bit odd. Do you mean “to verify that only N was limiting”?
- L. 84 Fig S1 suggests that for *Synechococcus*, adding Fe + NO₃⁻ has more effect than adding NO₃⁻ only. Do you think there is a co-limitation?
- L. 86-87 “Throughout the observation”: Table S1 in ref 14 only shows the initial nutrient concentrations, so I’m not sure what “Throughout the observation period” refers to. Or maybe the authors meant Fig S1 in ref 14? This can also be seen in Fig S2 of this manuscript.
- L. 93-94: the results in Table S2 show that in many replicates, urea is at least as good, and sometimes better than NH₄⁺ to stimulate *Prochlorococcus* growth (and similarly, sometimes as good or better than NO₃ for *Synechococcus*). I think this should be described and commented in the results, to explain why it was not included in the model (maybe because it was non-discriminant between the two organisms?).
- L. 108-123: The whole paragraph is hard to follow and needs rephrasing. Main ideas: it is energetically unexpected to prefer NO₃ over NH₄. This is not the first observation, but there is a lack of good explanation, both in *Synechococcus* and in other phytoplankton organisms.
- L. 132-133: it seems logical that when reversing the parameters that have been optimized for a given species, the fit is bad.
- L. 138-140: I am not sure to understand this sentence. Table S4 does not show other parameters, and the authors probably mean Table S3. But even in Table S3, only the mortality rates seem somewhat similar between the organisms (the quotas, v_{max} and K are all different)
- L. 140-142: I do not understand this sentence. I think the idea is that the differences in v_{max} are negligible compared to differences in K (in terms of their effect on the model output), which I agree with, but the sentence needs rephrasing. The following sentence (about affinity) expresses this idea well.
- L. 143 I would add the description of this measure of “affinity”, as described in ref 31: “a better indicator of nutrient uptake efficiency than Ks”
- L. 147-148: out of curiosity, have some K values been measured from culture experiments in the literature for these organisms?
- L. 152 As mentioned above, I’m not sure to understand the rationale behind the variation of K ratios. In addition, why are the solutions where one organism outcompetes the other considered as invalid?

- L. 163 Differential mortality could also be due to virus predation.
- L. 179-181 I am not sure to understand why the use of amino-acids by *Prochlorococcus* is discussed at this point of the manuscript. But I agree that the use of other sources of N (amino-acid, cyanate, urea...) by both *Prochlorococcus* and *Synechococcus* should be discussed in more depth, especially since their genomic potential for N assimilation is well described.
- L. 195-197 This sentence needs a reference.
- L. 208-210 This difference (and the variability within *Prochlorococcus*) has been known for a long time. Please cite the literature accordingly (e.g Moore *et al.* 2002, Martiny *et al.* 2009, Scanlan 2009, Partensky and Garczarek, 2010)
- L. 323: why were only NO₃⁻ and NH₄⁺ chosen to be part of the model (and not urea, for which data was also available?)
- L. 346-347: this is a bit unclear, maybe change to “For Fig. 3A, 4C, S5A, S5B, we varied the ratios of half-saturation constants, using NO₃⁻:NH₄⁺ resource ratios of 1:3, 2:3, 1:6 and 3:1, respectively”
- L. 348-354 This reads more like a discussion, and should to my opinion be included in the main text.
- L. 396 “For N and P addition, data is based on Exp. M1-M3, where the initial concentrations of N were small”: since the initial concentrations of N are used to select which data are represented on the figure, as well as for parametrizing the model (e.g. N quotas per cell), I believe that table S1 in ref 14 should be included in this manuscript as a supplemental table.
- L. 398 “Note that added NH₄⁺ was depleted on the third day”: again, this is an important information to interpret the results, and I believe that Fig S1 in ref 14 should be included in this manuscript as well. Or otherwise refer to Fig S2.
- L.411-412 Please explain in the legend (and maybe in the main text as well) why a ratio of 1:3 was used as a reference.
- Fig S1 legend “*Prochlorococcus* prefers NH₄⁺ while, *Synechococcus* prefers NO₃⁻.” From what I see, I would include urea in this sentence. In addition, I don’t think that these incubations tested for preference, but rather that *Prochlorococcus* was more efficient in growing on NH₄⁺ and urea, and *Synechococcus* more efficient in growing on NO₃⁻ (and urea). Due to transcriptomic regulation, if both were present at the same time, I am not sure that *Synechococcus* would prefer NO₃ over NH₄.

Typos

- L. 130: Table S6 does not show K values. The authors probably meant Table S3.
- L. 133-134 “On the other hand, K for NO₃⁻”: this is probably a typo to be removed.
- L. 141: it is table S3, not S5.
- L. 144: Table S4, not S5.
- L. 351 “and model the model”: typo.
- L. 403 Points in figure 2 are huge (in *Prochlorococcus*, they are almost 10000 cells/mL wide). I would suggest to reduce point size.
- L. 411-412 “The model output is based on the resource ratio of NO₃⁻ and NH₄⁺ is 1:3 (for other ratios, see Figure 3, S5).” There are 2 typos here (one extra “is”, and Figure 3 is mentioned in its own legend!).

- Table S2: typo in M1, day3, *Prochlorococcus*: (Cont = PO4) < (NO3 = Urea) < Urea. Typo for Fe1, Fe2, Fe3 for Day1 in *Prochlorococcus* and *Synechococcus*: includes NO3 and NH4 and Urea instead of Fe and PO4.

Response to comments

Please see the comments of reviewer below. As editor, I would like to see the diversity within *Prochlorococcus* and *Synechococcus* being explored and discussed. This impacts the conclusion of the study, as the results observed in the Northwestern Pacific cannot be generalized to other environmental context.

The main caveat is that the diversity within *Prochlorococcus* and *Synechococcus* is not discussed. In my opinion, this impacts the conclusion of the study, as the results observed in the Northwestern Pacific cannot be generalized to other environmental context. This diversity aspect is not even mentioned in the manuscript.

We appreciate this suggestion. We have already performed a phylogenomic analysis to address the issue of internal taxonomic and metabolic diversity within marine *Synechococcus* and *Prochlorococcus*, however, we did not include this thoroughly in the previous version of the main text for the sake of length. In the revised version of the manuscript we added an entire new section “**Different uptake affinity to N in the context of picocyanobacterial evolution**” to the manuscript discussing this topic, accompanied by a phylogenomic tree of picocyanobacteria (Fig. 6).

“Marine *Synechococcus* and *Prochlorococcus* are estimated to have sequentially diverged from a common ancestor during the Devonian, at about 413 million years ago (Mya), and Carboniferous, at around 360 Mya and 341 Mya, respectively (Sánchez-Baracaldo et al., 2019). During these periods, a number of genus-level lineages likely evolved within both *Synechococcus* (eg. *Parasynechococcus*, “*Pseudosynechococcus*”) and *Prochlorococcus* (eg. “*Thaumococcus*”, “*Prolificoccus*”, “*Riococcus*”, “*Eurycolium*”) (Komárek et al., 2020, Tschoeke et al., 2020, Salazar et al., 2020). Our reconstruction suggests that many other putative genera are still waiting to be recognised (Fig. 6). These lineages are in many aspects cryptic due to morphological simplification and genome streamlining, driven by convergent selective pressures, especially the nutrient deficiency addressed in this study (Partensky, 2010). Nevertheless, individual ecotypes and even individual cells of picocyanobacteria harbor distinct sets of metabolic genes reflecting their ecological niche partitioning (Kashtan et al., 2014, Biller et al., 2015., Doré et al., 2020).

The ability to assimilate NO_3^- is observed to vary widely among picocyanobacteria. Whereas most genomes of marine *Synechococcus* harbor both nitrate reductase and nitrite reductase genes, these genes exhibit a patchy distribution throughout both low-light and high-light adapted *Prochlorococcus* lineages (Casey et al., 2007, Berube et al., 2019) (Fig. 6). According to previous studies, gene loss, as well as homologous recombination among closely related cell populations, are frequent in the genomic regions responsible for NO_3^- acquisition in *Prochlorococcus* (Berube et al., 2019). Nitrate assimilation is energetically demanding and thus more likely to occur at high-light conditions near the water surface, but the concentration of NO_3^- tends to increase with depth in oligotrophic oceans. This trade-off likely shapes the composition of *Prochlorococcus* populations by creating an equilibrium of NO_3^- assimilating and non-assimilating sub-populations that co-exist at a given depth and light intensity to maximize the effectivity of N uptake (Berube et al., 2019).

Phylogenomic analysis (Fig. 6) implies that mixed populations of NO_3^- assimilating or non-assimilating “*Prochlorococcus*” cells corresponding to several different genus-level lineages may further influence the relative affinity of *Prochlorococcus* to different N substrates. This could be another mechanism underlying the co-existence of *Prochlorococcus* and *Synechococcus*, and their various phylogenetic sub-lineages, at different concentrations of available N forms. The relative contribution of genomic population diversity versus

physiological affinity to specific forms of N to the co-existence of both picocyanobacteria was not addressed in the current study, and thus warrants further investigation. With regard to the differential phylogeographic distribution of individual picocyanobacterial geno- and ecotypes (Zwirgmaier et al., 2008., Doré et al., 2022), care must be taken when extrapolating the findings to oceanic regions outside Northwest Pacific, in which verification of our hypothesis is still required.” (Lines 271-305)

Fig. 6. Phylogenomic tree of marine picocyanobacteria. The tree is rooted by the freshwater core cluster of *Synechococcus* (type strain *S. elongatus* PCC 6301). Individual clades are annotated following the most recent taxonomic revisions. a, b, c in superscript in the figure represent refs. (Komárek et al., 2020, Tschöeke et al., 2020, Salazar et al., 2020) respectively. Phylogenetic lineages are further classified using ecological groups of *Synechococcus* and *Prochlorococcus* (right); LL and HL stands for low-light and high-light adapted sub-clades of *Prochlorococcus* (Berube et al., 2015). The multi-genus clade traditionally considered as

Prochlorococcus sensu lato is highlighted in cyan. Presence of genes encoding nitrate reductase (*narB*) and nitrite reductase (*nirA*) was assessed for the individual ecological groups and expressed using the total number of occurrences within group along with a relative abundance scale bar. The tree is based on 120 conserved bacterial proteins using the Approximately Maximum Likelihood method following the GTDB-Tk *de novo* pipeline (Chaumeil et al., 2019). Bootstrap values are shown near the nodes, numbers in brackets indicate the number of genomes within collapsed clades.

Related references---

Sánchez-Baracaldo P, Bianchini G, Di Cesare A, Callieri C, Christmas NAM. 2019. Insights into the evolution of picocyanobacteria and phycoerythrin genes (*mpeBA* and *cpeBA*). *Front Microbiol* 10:45

Komárek J, Johansen JR, Šmarda J, Strunecký O. 2020. Phylogeny and taxonomy of *Synechococcus*-like cyanobacteria. *Fottea* 20:171-191.

Tschoeke D, Salazar VW, Vidal L, Campeão M, Swings J, Thompson F, Thompson C. 2020. Unlocking the genomic taxonomy of the *Prochlorococcus* collective. *Microb Ecol* 80:546-558.

Salazar VW, Tschoeke DA, Swings J, Cosenza CA, Mattoso M, Thompson CC, Thompson FL. 2020. A new genomic taxonomy system for the *Synechococcus* collective. *Environ Microbiol* 22:4557-4570

Partensky F, Garczarek L. 2010. *Prochlorococcus*: advantages and limits of minimalism. *Ann Rev Mar Sci* 2:305-31.

Kashtan N, Roggensack SE, Rodrigue S, Thompson JW, Biller SJ, Coe A, Ding H, Martinen P, Malmstrom RR, Stocker R, Follows MJ, Stepanauskas R, Chisholm SW. 2014. Single-cell genomics reveals hundreds of coexisting subpopulations in wild *Prochlorococcus*. *Science* 344:416-20.

Biller SJ, Berube PM, Lindell D, Chisholm SW. 2015. *Prochlorococcus*: the structure and function of collective diversity. *Nat Rev Microbiol* 13:13-27.

Doré H, Farrant GK, Guyet U, Haguait J, Humily F, Ratin M, Pitt FD, Ostrowski M, Six C, Brillet-Guéguen L, Hoebeke M, Bisch A, Le Corguillé G, Corre E, Labadie K, Aury JM, Wincker P, Choi DH, Noh JH, Eveillard D, Scanlan DJ, Partensky F, Garczarek L. 2020. Evolutionary Mechanisms of Long-Term Genome Diversification Associated With Niche Partitioning in Marine Picocyanobacteria. *Front Microbiol* 11:567431.

Casey JR, Lomas MW, Mandecki J, Walker DE. 2007. *Prochlorococcus* contributes to new production in the Sargasso Sea deep chlorophyll maximum. *Geophys Res Lett* 34.

Berube PM, Rasmussen A, Braakman R, Stepanauskas R, Chisholm SW. 2019. Emergence of trait variability through the lens of nitrogen assimilation in *Prochlorococcus*. *Elife* 8.

Zwirgmaier K, Jardillier L, Ostrowski M, Mazard S, Garczarek L, Vaultot D, Not F, Massana R, Ulloa O, Scanlan DJ. 2008. Global phylogeography of marine *Synechococcus* and *Prochlorococcus* reveals a distinct partitioning of lineages among oceanic biomes. *Environ Microbiol* 10:147-61.

Doré H, Leconte J, Guyet U, Breton S, Farrant GK, Demory D, Ratin M, Hoebeke M, Corre E, Pitt FD, Ostrowski M, Scanlan DJ, Partensky F, Six C, Garczareka L. 2022. Global phylogeography of marine *Synechococcus* in coastal areas reveals strong community shifts. *mSystems* 7:10.1128/msystems.00656-22.

Berube PM, Biller SJ, Kent AG, Berta-Thompson JW, Roggensack SE, Roache-Johnson KH, Ackerman M, Moore LR, Meisel JD, Sher D, Thompson LR, Campbell L, Martiny AC, Chisholm SW. 2015. Physiology and evolution of nitrate acquisition in *Prochlorococcus*. *ISME J* 9:1195-1207.

Chaumeil PA, Mussig AJ, Hugenholtz P, Parks DH. 2019. GTDB-Tk: a toolkit to classify genomes with the Genome Taxonomy Database. *Bioinformatics* doi:10.1093/bioinformatics/btz848.

Additional details are described as a response to the reviewer's comments below.

Reviewer #1 (Comments for the Author):

Overall, I found the paper to be well written. It was clearly structured and therefore easy to follow through. The objectives of the study were clearly listed in the introduction and answered in the results and discussion. The data presentation and use of statistics was suitable and useful to understand the data collected. I also thought that the discussion was well supported with previous studies. I especially found the use of phylogenetic analysis of genes involved in nitrate utilization to be useful supporting information.

We appreciate the time and effort devoted to reviewing and polishing our work. We revised the manuscript following the comments. We describe our response to each point below:

I have two major comments.

- Would there be any niche partitioning by depth between *Prochlorococcus* and *Synechococcus*, as the concentration of nitrate is lower at the surface and increases with depth? Are there any comparisons between cell concentration of *Prochlorococcus* and *Synechococcus* at the deep chlorophyll maximum and correlating to the relative concentrations of nitrate to ammonia?

The earlier review by Partensky (Partensky et al., 1999) described the niche partitioning of *Prochlorococcus* and *Synechococcus* by depth. However, it doesn't explain the preference for NO_3^- by *Synechococcus*. They described that in oligotrophic waters when nitracline depth is located at the bottom of the euphotic layer, there is a marked *Prochlorococcus* abundance peak just above the nitracline and much fewer cells in the surface waters, while for *Synechococcus*, the abundance peak is low all along the vertical profile and the abundance peak is reduced (Olson et al., 1990, Campbell et al., 1993). We believe that this discrepancy suggests considering the nutrient history of phytoplankton: frequency, scale and duration of nutrient input (Droop, 1973).

" The preference of NO_3^- by *Synechococcus* cannot be explained by the snapshot of vertical distribution in oligotrophic waters (Partensky et al., 1999) being a marked *Prochlorococcus* abundance peak just above the nitracline, while for *Synechococcus*, the abundance peak is low all along the vertical profile (Olson et al., 1990, Campbell et al., 1993). This suggests us to consider the nutrient history of phytoplankton: frequency, scale and duration of nutrient input (Droop, 1973). "(Lines 222-227)

In this study, we harvested water samples from 10 m depth to monitor *Prochlorococcus* and *Synechococcus*, which are supported by recycled nitrogen without light limitations.

Related references---

Partensky F., et al. (1999). Differential distribution and ecology of *Prochlorococcus* and *Synechococcus* in oceanic waters: a review." *Bull Inst Oceanogr* 19: 457-475.

Olson et al., (1990). Spatial and temporal distributions of prochlorophyte picoplankton in the North Atlantic Ocean. *Deep Sea Res.* 37, 1033-1051.

Campbell L., Vaclot D., (1993) Photosynthetic picoplankton community structure in the subtropical North Pacific Ocean near Hawaii (station ALOHA). *Deep Sea Res.*, 40, 2043-2060.

Droop MR. 1973. Some thoughts on nutrient limitation in algae. J Phycol 9:264 - 272.

- Please provide some references for line 197 stating that ocean stratification reduces supply of nitrate to photic zone of low latitudes. What is the predicted magnitude of decrease nitrate and how does it relate to the concentrations of nitrate and ammonia used in this study?

We thank the reviewer's comment. We compiled climate simulation and plotted shift in NO_3^- in low latitude euphotic zones (30S-30N, 0-50m), which is new panel in Fig 5 (Fig. 5D). The results are consistent in declining NO_3^- (on average ~40%), indicating substantial decline in NO_3^- sources. We included a reference (Taylor et al., 2012) in the new method section describing this analysis (Lines 464-471). Our study is based on *in-situ* concentrations at nanomolar levels. Global models tend to be calibrated with micromolar concentrations and thus may require additional calibration, which is beyond the scope of this study.

Fig. 5D NO_3^- concentration shifts predicted by various climate models (See methods). The y-axis represents the fractional (%) change in surface nutrient concentration throughout the low latitudes (30S-30N, 0-50m). For models without NO_3^- , the shifts in NO_3^- are estimated from PO_4^{3-} with a typical $\text{NO}_3^-:\text{PO}_4^{3-}$ ratio of 15.

Since *Synechococcus* can utilize nitrate, as proven from culture based studies, the magnitude of change might not be as large as suggested from the data.

We appreciate the reviewer's comment. *Synechococcus* utilizing nitrate is consistent with our model setting. In case the reviewer meant *Prochlorococcus*'s utilizing nitrate, it is reflected in the model as well, yet with lower efficiency than *Synechococcus*. [eq. 1, 2] Table S5.

I also have a few minor comments.

- Fig 1 - what does the dotted line represent?

Thank you for pointing out the mistake. The dotted line presents the value of control (=1). We included the sentence in the caption.

Caption for Fig. 1: "The effects of different chemical forms of N, P and Fe additions on cell abundances of *Prochlorococcus* (A) and *Synechococcus* (B) relative to controls (means \pm SD), with a number of samples in parentheses. For N and P addition, data is based on Exp. M1-M3, where the initial concentrations of N were small (<10 nM) (Table S2). It shows that

Prochlorococcus prefers NH_4^+ while *Synechococcus* prefers NO_3^- . Note that added NH_4^+ was depleted on the third day, leading to decreased populations in both organisms (Fig. S2). The dotted line presents the value of control."

- Fig 2 - are the bars around the data points representing standard error?

These represent standard deviation based on the representative values of Ex1-3. We now have clarified this in the figure caption.

Caption for Fig. 2 "Model data comparison of the time series of abundance (X) of *Prochlorococcus* and *Synechococcus*. (A) NH_4^+ added. (B) NO_3^- added. Points, Data; Curves, Model. Pro, *Prochlorococcus*; Syn, *Synechococcus*. Data are based on Exp. M1-M3, where the initial concentrations of N were small (<10 nM) (Table S2). The error bars represent the standard deviation based on the mean values across from the selected experiments."

- Fig 3 and 4 - resolution of the figures are too low to read the axis.

We have improved the resolution of these figures (New Fig. 4 and 5). To ensure readability, we now have a bigger font and tick sizes for the axes of these figures.

- The manuscript has a few informal language use eg. "can't". Please change to formal language. "can't" was replaced with "cannot" (Lines 200, 223)

- Line 125 - is there another subheading that you could give this section for it to be more representative? One suggestion I can make is "Ecological model for N uptake preferences" We appreciate the reviewer for the great suggestion. We now use the suggested title for this section (Line 120).

- Line 134 - There seems to be formatting error for the sentence "K for NO_3^- -Similarly..." We apologize for the error. The now reads: "On the other hand, K for NO_3^- (K_{NO_3}) must be lower for *Synechococcus* to reproduce its continued high growth up to day 3 (Fig. 2B) and reversal of species NO_3^- affinities again results in a poor model fit to data (Fig. S5)" (Lines 136-138)

- I may have missed this in the text, but what is the pore size of the filter used to prefilter seawater samples to reduce grazing pressure?

We used 1 μm pore size of the filter for macro-nutrient enrichment experiments (M1 to M5) and 10 μm pore size for Fe enrichment experiments (Fe1 to Fe3) to reduce grazing pressure.

"For the bioassay treatments with N and P addition, water was filtered through an acid-cleaned 1 μm in-line cartridge filter (Micropore EU, ORGANO). For Fe additions, seawater was pre-filtered through a 10 μm filter of the same manufacturer." (Lines 358-360)

Reviewer 2

In their article Coexistence of dominant marine phytoplankton sustained by nutrient specialization, Masuda and co-authors explore a potential hypothesis to explain the coexistence of the two genera of marine picocyanobacteria *Synechococcus* and *Prochlorococcus* in the the subtropical Northwestern Pacific. Through a set of on-deck incubation, they show that *Prochlorococcus* grows faster after NH_4^+ addition than after NO_3^- addition, while the opposite pattern is observed for *Synechococcus*. Based on these incubation results, they then fit a simple growth model and estimate the nitrogen assimilation parameters associated to each genus. These parameters are used to discuss what allows for the coexistence of the two genera. The main result is that coexistence is permitted by different affinities of *Prochlorococcus* and *Synechococcus* to nitrate and nitrite.

While it is long known that *Prochlorococcus* and *Synechococcus* can co-occur, and that *Prochlorococcus* dominates in oligotrophic waters and *Synechococcus* in richer waters, to my knowledge no study has tried to find a mechanistic explanation for their coexistence. The two organisms and their physiologies are so different that it is probably a bit simplistic to reduce their coexistence to the use of different forms of nitrogen, but it is nonetheless interesting to highlight their difference in this regard. While the description of the results of on-deck incubations is relatively clear, the reasoning of the authors is sometimes hard to follow in the description of the model and its outputs, and the discussion fails to take into account the current knowledge of *Prochlorococcus* and *Synechococcus* diversity. These aspects are further developed in my comments below.

We appreciate the time and effort devoted to carefully reviewing and constructive comments to polish our work. We revised the manuscript following the comments. We describe our response to each point below:

Main comments:

- Throughout the manuscript, the authors refer to *Synechococcus* and *Prochlorococcus* as different species, and explore the question of species coexistence. While the taxonomy of cyanobacteria still needs revision, neither *Synechococcus* nor *Prochlorococcus* can be considered as species-level taxonomic groups according to current genomic standard (see e.g. Salazar et al. 2020, Tschoeke et al. 2020, Doré et al. 2020), and they should be referred to as different genera. Both genera encompass a wide diversity described by an extensive literature, and the variability N acquisition abilities within each genus is well documented both by culture experiments (e.g. Moore et al 2002) and genomic analyses (e.g. Scanlan et al. 2009, Martiny et al 2009, Partensky and Garczarek 2010, Doré et al 2020). As a consequence, the results obtained with on-deck incubations are only valid for the community present at the sampling sites, and cannot be generalized to other environmental contexts. Different strains of *Synechococcus* and *Prochlorococcus* most likely have different half-saturation constants (K) for NO_3^- and NH_4^+ . The strain variability should thus at least be discussed somewhere in the manuscript.

We agree that our study is at genus-level, but not species-level, and these terms were revised as suggested (Lines 44, 46, 49, 50, 70, 278). We also agree with the reviewer's suggestion to discuss the strain level diversity and we have added a whole new section "**Different uptake affinity to N in the content of picocyanobacterial evolution**" which includes an extended discussion about such diversity (Lines 271 - 305, and Lines 473 - 487 for its methods). The discussion includes reference to several studies suggested by the reviewer, accompanied by a phylogenomic tree of picocyanobacteria (Fig. 6), discussed in the context of this study. We

also acknowledged (in the revised version) the limits of extrapolating our findings outside the study area, taking into consideration the phylogeographic patterns of picocyanobacterial geno- and ecotypes (e.g. Doré et al. 2022).

- I don't understand the rationale behind the variation of K_{pro}/K_{syn} in Figures 3, 4 and S5. I understand that only some pairs of K_{pro} and K_{syn} for NO_3 and NH_4 allow for their coexistence, but these parameters have been determined using the model fit to the data, so why having them vary? What does that represent biologically?

I find more interesting to see in which $NO_3:NH_4$ ratios *Synechococcus* and *Prochlorococcus* can coexist, given the parameters estimated by the model, as shown in Fig 4A.

We appreciate the reviewer's comment and your interest in the new Fig. 5A. These simulations with K_{pro}/K_{syn} have several purposes. The first purpose is to show that there are large areas where the co-existence may not occur and thus, K_{pro}/K_{syn} must be at a certain ratio. The optimized K_{pro}/K_{syn} value allows the co-existence because the triangle in Fig. 4 is in the coexistence range. Second, the figure shows how the model works by taking the different dimensions of axes than Fig. 5A. The change in K_{pro}/K_{syn} has an impact on whether coexistence may occur or not, and also what type of co-existence if it occurs (i.e., whether *Prochlorococcus* use NH_4^+ or NO_3^-). The model also shows how the area of coexistence may shift depending on the NH_4^+ and NO_3^- resource ratios, providing an explanation behind Fig. 5A.

Regarding "these parameters have been determined using the model fit to the data, so why having them vary?" we vary them to show that the optimized K values for these organisms allow these organisms to co-exist, by showing that other K value combinations may not allow it, supporting the mechanism that allows their coexistence.

We consider the triangle in Fig. 4 an actual value and other combinations of K values are what-if scenarios, and the model output is to explore how K values (or uptake properties) may affect whether the coexistence or one-genus-domination may occur.

Overall these exercises are done to show how K values matter regarding the co-existence of these organisms, as well as the resource ratio of NH_4^+ and NO_3^- .

We added a sentence to convey this point:

"The result shows that these K value ratios strongly influence the co-existence of these organisms at a given resource ratios of NO_3^- and NH_4^+ ." (Lines 168-169)

Regarding "What does that represent biologically?", the differentiating K values for *Prochlorococcus* and *Synechococcus* allow their coexistence, with higher uptake of *Synechococcus* mainly supported by NO_3^- and *Prochlorococcus* NH_4^+ and they evolved in such a way to differentiate N uptakes.

Also, one probably most supported view is that the K value represents the density of the transporter at the cell surface (Armstrong, 2008, Smith et al., 2009, Bonachela et al., 2011). NH_4^+ and NO_3^- are ionized, thus require transporters to be taken into the cell, which is different than small non-ionized molecules like CO_2 , N_2 , O_2 , which may go through the cell membrane. High density of transporters (represented by low K value) allows lowering the concentration of NH_4^+ and NO_3^- at the boundary layer allowing high efficient transport/uptake of nutrient even where its concentration is low.

We have added this idea in the main text.

“K values have been thought to represent the density of transporters on the cellular surface (Armstrong, 2008, Smith et al., 2009, Bonachela et al., 2011), with lower value representing a higher transporter density.” (Lines 131-133)

Related references---

Doré H, Leconte J, Guyet U, Breton S, Farrant GK, Demory D, Ratin M, Hoebeke M, Corre E, Pitt FD, Ostrowski M, Scanlan DJ, Partensky F, Six C, Garczareka L. 2022. Global phylogeography of marine *Synechococcus* in coastal areas reveals strong community shifts. *mSystems* 7:10.1128/msystems.00656-22.

Armstrong RA, 2008. Nutrient uptake rate as a function of cell size and surface transporter density: A Michaelis-like approximation to the model of Pasciak and Gavis. *Deep Sea Res. I.*, 55, 1311-1317.

Smith SL, Yamanaka Y, Pahlow M, Oschlies A. 2009. Optimal uptake kinetics: physiological acclimation explains the pattern of nitrate uptake by phytoplankton in the ocean. *Mar Ecol Prog Ser* 384:1-12.

Bonachela JA., Raghiv M and Levin SA. 2011. Dynamic model of flexible phytoplankton nutrient uptake. *Proc Natl Acad Sci U S A* 108:20633-8.

Line by line comments:

- L.39 "higher preference": could be more specific by saying "higher affinity" since the difference is in K. In addition, the authors did not test for the preference (which would mean to grow the bacteria with both nutrients, and see which one is used) but for the efficiency of growth with each nutrient. Due to transcriptomic regulation, if both were present at the same time, I am not sure that *Synechococcus* would prefer NO₃ over NH₄.

We agree with the reviewer's suggestion and revised the term "preference" to "affinity" (Line 42).

- L. 41-42: "is sufficient to explain their coexistence". I'm not sure that we can say it is sufficient. The model demonstrates that coexistence is possible when they have different nutrient affinities.

We appreciate the reviewer's comments. We have now toned down the statement.

“A simple ecological model demonstrates that the differential nutrient preference inferred from field experiments with these genera is capable of sustaining their coexistence.” (Lines 43-45)

- L. 45 "species": as mentioned above, *Synechococcus* and *Prochlorococcus* are different genera. I would suggest to replace species by genera here and in the rest of the manuscript when referring to *Prochlorococcus* and *Synechococcus*.

Thank you for the advice, we revised the term "species" by "genera" (Lines 44, 46, 49, 50).

- L. 57 "Nutrient utilization is a critical factor in microbial co-existence." It would be nice to have a citation from literature for this affirmation.

We refer to Hutchinson et al., 1961, Tilman 1977 and Ward et al., 2013 to support the sentence. "Nutrient utilization is a critical factor in microbial co-existence (Hutchinson, 1961, Tilman, 1977, Ward et al., 2013) (Line 61)"

Related reference---

Hutchinson GE. 1961. The Paradox of the Plankton. *The American Naturalist* 95:137-145.

Tilman D. 1977. Resource competition between plankton algae: An experimental and theoretical approach. *Ecology* 8:338-348.

Ward BA, Dutkiewicz S, Moore CM, Follows MJ. 2013. Iron, phosphorus, and nitrogen supply ratios define the biogeography of nitrogen fixation. *Limnol Oceanogr* 58:2059-2075.

- L.64 Reference 7 (Martiny et al., 2009) concerns *Prochlorococcus* (and *Synechococcus*). This would be clearer if it was stated in the sentence, as the previous sentences are much more general.

We specified the reference (Martiny et al., 2009) concerns *Prochlorococcus* and *Synechococcus*: "Qualitative evidence supports differential nitrogen utilization in *Prochlorococcus* and *Synechococcus* (Martiny et al., 2009), but the traits and conditions that favor co-existence or dominance of these genera have not yet been quantitatively demonstrated. (Lines 68-69)".

Related reference---

Martiny AC, Kathuria S, Berube PM. 2009. Widespread metabolic potential for nitrite and nitrate assimilation among *Prochlorococcus* ecotypes. *Proc Natl Acad Sci U S A* 106:10787–10792.

- L. 83-84 The sentence reads a bit odd. Do you mean "to verify that only N was limiting"? We corrected it following your suggestion.(Line 89-90)

- L. 84 Fig S1 suggests that for *Synechococcus*, adding Fe + NO₃⁻ has more effect than adding NO₃⁻ only. Do you think there is a co-limitation?

We appreciate the question. We do not think *Synechococcus*'s growth was explicitly co-limited by Fe and NO₃⁻, since cell abundances on day 3 were similar: about 3000 cells/mL for both NO₃⁻ and Fe + NO₃⁻ (Fig. S1B and D). Also, on the 5th day of the Fe added cases, only the one with NO₃⁻ shows an exceptionally high value (compared to e.g., Fe only or Fe + PO₄³⁻), suggesting that nitrogen is the dominant growth limiting factor.

- L. 86-87 "Throughout the observation": Table S1 in ref 14 only shows the initial nutrient concentrations, so I'm not sure what "Throughout the observation period" refers to. Or maybe the authors meant Fig S1 in ref 14? This can also be seen in Fig S2 of this manuscript.

We are sorry for the confusion. We revised the sentence to clarify the message that the initial nutrient concentration was low for all the nutrient amendment experiments: "Initial nutrient concentrations remained low during the cruise (<40 nM for N ~30 nM for P and more than 0.05 nM of dissolved iron. Table S2)". (Lines 93-94)

- L. 93-94: the results in Table S2 show that in many replicates, urea is at least as good, and sometimes better than NH₄⁺ to stimulate *Prochlorococcus* growth (and similarly, sometimes as good or better than NO₃ for *Synechococcus*). I think this should be described and commented in the results, to explain why it was not included in the model (maybe because it was non-discriminant between the two organisms?).

We agree with the reviewer's opinion that urea stimulates growth at least as good as ammonium or nitrate for *Prochlorococcus* and *Synechococcus*, respectively. We selected NH_4^+ and NO_3^- because of consistent and different trends between these two organisms. While we didn't select urea for modelling because of its non-discriminant results between *Prochlorococcus* and *Synechococcus*. We described these in the current version:

" Urea stimulated the growth of *Prochlorococcus* and *Synechococcus* similar to or sometimes better than ammonium or nitrate, respectively (Fig. S1C and D, Table S3). " (Lines 101-103)

"We have selected NO_3^- and NH_4^+ to be part of the model mainly to focus on delivering the most striking finding extracted from the data - the different affinity to chemical forms of inorganic nitrogen between two organisms. Also, inorganic nutrients are more commonly used in the ecological modelling and we have followed the previous similar studies focusing on inorganic nutrients as a starter (Tilman, 1977, Ward et al., 2013)."(Lines 124-128)

Related references---

Tilman, D. 1977. Resource competition between plankton algae: An experimental and theoretical approach." Ecology 8: 338-348.

Ward, B. A., et al. (2013). "Iron, phosphorus, and nitrogen supply ratios define the biogeography of nitrogen fixation." Limnology and Oceanography 58(6): 2059-2075.

- L. 108-123: The whole paragraph is hard to follow and needs rephrasing. Main ideas: it is energetically unexpected to prefer NO_3^- over NH_4^+ . This is not the first observation, but there is a lack of good explanation, both in *Synechococcus* and in other phytoplankton organisms.

We appreciate the reviewer's comments. We believe that the key idea here is that at high concentration, the result follows what is expected based on the energetics, but at low concentration for *Synechococcus*, it is not the case. To clarify the story, we added new Table 1 and Fig. 3, revised and partly moved the sentences as following:

"The model parametrization suggests higher maximum uptake (V_{max}) of NO_3^- by *Synechococcus* than *Prochlorococcus* (Table S4), which also may contribute to the niche differentiation. It also indicates that V_{max} for NH_4^+ is higher than NO_3^- both for *Prochlorococcus* and *Synechococcus*. Thus, under high nutrient concentrations, *Synechococcus* may grow faster with NH_4^+ than NO_3^- (Table 1, Fig 3). This model implication is consistent with the previous experiments where high amount (micro molar level) of NH_4^+ and NO_3^- is added (Moore et al., 2002, Glibert and Ray, 1990). However, the affinity ($A = V_{max}/K$) for NO_3^- turned out to be higher than that for NH_4^+ for *Synechococcus* (Table S5). The affinity provides an alternative measure of the relative ability of various species to compete for nutrients (Smith et al., 2009, Healey, 1980). Because the affinity is the initial slope of nutrient uptake vs nutrient concentration relationship (Van de Waal and Litchman, 2020), the value of affinity is especially relevant when nutrients are depleted as in the subtropical gyres, where small phytoplankton tend to dominate (Hirata et al., 2011). The result shows *Synechococcus* has an advantage for NO_3^- and *Prochlorococcus* has an advantage for NH_4^+ (Table S5), consistent with what is suggested by the predicted half-saturation values (and not by maximum uptake rates). These results collectively suggest that in low nutrient environment as in the subtropical gyres, these organisms have different nitrogen uptake preferences for NH_4^+ and

NO_3^- (Table 1, Fig 3), allowing their co-existence, and the differentiated half-saturation constants are the key contributors. " (Lines 142-159)

	Low nutrient*	High nutrient*
Prochlorococcus	NH_4^+	NH_4^+
Synechococcus	NO_3^-	NH_4^+

*Broadly, low nutrient indicates lower nano-molar level (e.g., surface of subtropical gyres (this study)) and high nutrient indicates above that level (initial conditions of batch culture experiments (Moore et al., 2002, Glibert and Ray 1990, Kudo and Harrison, 1997). Specifically, the nutrient preference of *Synechococcus* is predicted to be flipped at 39 nM (Fig. 3).

Fig. 3 Nutrient uptake rates V_N against nutrient concentration $[N]$. In the legend *Pro* and *Syn* indicate *Prochlorococcus* and *Synechococcus* respectively and *NH4* and *NO3* indicate ammonium and nitrate respectively. For example, $V_{\text{NH}_4}^{\text{Pro}}$ indicates ammonium uptake rate by *Prochlorococcus* vs ammonium concentrations.

Related references---

Moore LR, Post AF, Rocap G, Chisholm SW. 2002. Utilization of different nitrogen sources by the marine cyanobacteria *Prochlorococcus* and *Synechococcus*. *Limnol Oceanogr* 47:989-996.

Glibert PM, Ray RT. 1990. Different patterns of growth and nitrogen uptake in two clones of marine *Synechococcus* spp. *Mar Biol* 107:273-280.

Smith SL, Yamanaka Y, Pahlow M, Oschlies A. 2009. Optimal uptake kinetics: physiological acclimation explains the pattern of nitrate uptake by phytoplankton in the ocean. *Mar Ecol Prog Ser* 384:1-12.

Healey FP. 1980. Slope of the Monod equation as an indicator of advantage in nutrient competition. *Microb Ecol* 5:281 - 286.

Van de Waal DB, Litchman E. 2020. Multiple global change stressor effects on phytoplankton nutrient acquisition in a future ocean. *Philos Trans R Soc Lond B Biol Sci* 375:20190706.

Hirata T, Hardman-Mountford NJ, Brewin RJW, Aiken J, Barlow R, Suzuki K, Isada T, Howell E, Hashioka T, Noguchi-Aita M, Yamanaka Y. 2011. Synoptic relationships between surface Chlorophyll-*a*; and diagnostic pigments specific to phytoplankton functional types. *Biogeosciences* 8:311-327.

Kudo I, Harrison. PJ. 1997. Effect of iron nutrition on the marine cyanobacterium *Synechococcus* grown on different N sources and irradiances. *J Phycol* 33:232–240.

- L. 132-133: it seems logical that when reversing the parameters that have been optimized for a given species, the fit is bad.

We appreciate the reviewer's comments. We wish to point out that there are cases that substantially different sets of parameters may similarly reproduce the data, and the point of the modeling exercise here is to show that this is not likely the case here.

- L. 138-140: I am not sure to understand this sentence. Table S4 does not show other parameters, and the authors probably mean Table S3. But even in Table S3, only the mortality rates seem somewhat similar between the organisms (the quotas, v_{max} and K are all different)

We apologize for the confusion. We meant previous Table S3, and current Table S4. Also, we meant that the difference of other parameters between *Prochlorococcus* and *Synechococcus* are smaller than that for K values. For example, $K_{NH_4}^{Pro}$ and $K_{NH_4}^{Syn}$ are about 6 times different, and such a large difference is not shown in other parameters between these organisms. We have revised the sentence to improve the clarity.

" A lower K_{NH_4} for *Prochlorococcus* was essential to reproduce the rapid *Prochlorococcus* growth up to day 2 and the slower *Synechococcus* growth with NH_4^+ addition (Fig. 2A); when the species' relative NH_4^+ affinities are reversed, the model results deviate from the data (Fig. S4). On the other hand, K for NO_3^- (K_{NO_3}) must be lower for *Synechococcus* to reproduce its continued high growth up to day 3 (Fig. 2B) and reversal of species NO_3^- affinities again results in a poor model fit to data (Fig. S5)." (Lines 133-138)

- L. 140-142: I do not understand this sentence. I think the idea is that the differences in v_{max} are negligible compared to differences in K (in terms of their effect on the model output), which I agree with, but the sentence needs rephrasing. The following sentence (about affinity) expresses this idea well.

We thank the reviewer's comment. We understand that the sentence was confusing. We now removed the sentence, as we thought that the following sentence was sufficient and agree with the reviewer that it is clearer.

- L. 143 I would add the description of this measure of "affinity", as described in ref 31: "a better indicator of nutrient uptake efficiency than K_s "

We appreciate the suggestion. We feel that whether the affinity is the better indicator for nutrient uptake efficiency may depend on the context. For example, as the nutrient concentration increases, the value of affinity becomes less relevant as it is the initial slope in

the uptake rate vs nutrient concentration curves. Thus we used the term “alternative” instead of “better” and added that the value is especially relevant under low nutrient:

"The affinity provides an alternative measure of the relative ability of various species to compete for nutrients (Smith et al., 1980, Van de Waal and Litchman, 2020). Because the affinity is the initial slope of nutrient uptake vs nutrient concentration relationship (Van de Waal and Litchman, 2020), the value of affinity is especially relevant when nutrients are depleted as in the subtropical gyres, where small phytoplankton tend to dominate (Hirata et al., 2011)." (Lines 149-153)

Related references---

Healey, F. P. (1980). "Slope of the Monod equation as an indicator of advantage in nutrient competition." *Microbial Ecology* 5: 281 - 286.

Smith SL, Yamanaka Y, Pahlow M, Oschlies A. 2009. Optimal uptake kinetics: physiological acclimation explains the pattern of nitrate uptake by phytoplankton in the ocean. *Marine Ecology Progress Series* 384:1-12

Van de Waal DB, Litchman E. 2020. Multiple global change stressor effects on phytoplankton nutrient acquisition in a future ocean. *Philos Trans R Soc Lond B Biol Sci* 375:20190706.

Hirata T, Hardman-Mountford NJ, Brewin RJW, Aiken J, Barlow R, Suzuki K, Isada T, Howell E, Hashioka T, Noguchi-Aita M, Yamanaka Y. 2011. Synoptic relationships between surface Chlorophyll-*a*; and diagnostic pigments specific to phytoplankton functional types. *Biogeosciences* 8:311-327.

- L. 147-148: out of curiosity, have some K values been measured from culture experiments in the literature for these organisms?

To the best of our knowledge, there are no K_s values reported from any culture experiments for both organisms. This may be largely because of the measurement accuracy (detection limit of 0.05 μM at the best condition) that other studies apply. In our study, we used a high accuracy method enabling nM level with detection limits of 3-7 nM measurement allowing us to estimate the K_s values.

- L. 152 As mentioned above, I'm not sure to understand the rationale behind the variation of K ratios. In addition, why are the solutions where one organism outcompetes the other considered as invalid?

For the first part, please see the response above. For the second part, the equations are based on the premise that co-existence occurs (e.g., there are uptake by the two different organisms in equations), and thus when one outcompetes others, the computer provides invalid numbers (nan: not a number). We removed this sentence to avoid confusion. Instead, we added the following explanation in methods:

“Because the equations are based on the existence of these two organisms, valid solutions are obtained only where these organisms coexist.” (Lines 449-450)

- L. 163 Differential mortality could also be due to virus predation.

We agree with the advice given and included the virus predation as a possible reason for differential mortality (Lines 180-182).

"Preferential grazing by zooplankton (e.g., nanoflagellates (Christaki et al., 2002)) or virus (Avrani et al., 2011, Marston et al., 2012) may further ensure the coexistence of these species (Vallina et al., 2014, Ward et al., 2014, Dutkiewicz et al., 2020, Follows et al., 2018), and this has been the main explanation for the coexistence of these organisms."

Related references---

Christaki U, Courties C, Karayanni H, Giannakourou A, Maravelias C, Kormas KA, Lebaron P. 2002. Dynamic characteristics of *Prochlorococcus* and *Synechococcus* consumption by bacterivorous nanoflagellates. *Microb Ecol* 43:341-52

Avrani S, Wurtzel O, Sharon I, Sorek R, Lindell D. 2011. Genomic island variability facilitates *Prochlorococcus*-virus coexistence. *Nature* 474:604-8.

Marston MF, Pierciey FJ, Jr., Shepard A, Gearin G, Qi J, Yandava C, Schuster SC, Henn MR, Martiny JB. 2012. Rapid diversification of coevolving marine *Synechococcus* and a virus. *Proc Natl Acad Sci U S A* 109:4544-9.

Vallina SM, Ward BA, Dutkiewicz S, Follows MJ. 2014. Maximal feeding with active prey-switching: A kill-the-winner functional response and its effect on global diversity and biogeography. *Prog Oceanogr* 120:93-109.

Ward BA, Dutkiewicz S, Follows MJ. 2014. Modelling spatial and temporal patterns in size-structured marine plankton communities: top-down and bottom-up controls. *J Plankton Res* 36:31-47.

Dutkiewicz S, Cermeno P, Jahn O, Follows MJ, Hickman AE, Taniguchi DAA, Ward BA. 2020. Dimensions of marine phytoplankton diversity. *Biogeosciences* 17:609-634.

Follows MJ, Dutkiewicz S, Ward BA, Follett CN. 2018. Theoretical interpretation of subtropical plankton biogeography. *Microbial Ecology of the Oceans*, edited by: Gasol, J and Kirshman, D, 3rd Edn, John Wiley, Hoboken, NJ, p 467.

- L. 179-181 I am not sure to understand why the use of amino-acids by *Prochlorococcus* is discussed at this point of the manuscript. But I agree that the use of other sources of N (amino-acid, cyanate, urea...) by both *Prochlorococcus* and *Synechococcus* should be discussed in more depth, especially since their genomic potential for N assimilation is well described.

Following the advice, we moved the discussion related to amino acids to the end of paragraph starting from "The niche partitioning of Pro/Syn based on differential preferences for reduced and oxidized forms of N has ecological and climatic implications" (Lines 233-245).

- L. 195-197 This sentence needs a reference.

Following a reviewer, we now added a reference. (Line214)

Related reference---

Bograd SJ, Jacox MG, Hazen EL, Lovecchio E, Montes I, Pozo Buil M, Shannon LJ, Sydeman WJ, Rykaczewski RR. 2023. Climate change impacts on eastern boundary upwelling systems. *Ann Rev Mar Sci* 15:303-328.

- L. 208-210 This difference (and the variability within *Prochlorococcus*) has been known for a long time. Please cite the literature accordingly (e.g Moore et al. 2002, Martiny et al. 2009, Scanlan 2009, Partensky and Garczarek, 2010)

We appreciate the suggestion and included related references.

“The predicted and observed preference of NO_3^- by *Synechococcus* is qualitatively consistent with *phylogenetic* analysis, where *Synechococcus* possess genes encoding both nitrate reductase (*narB*) and nitrite reductase (*nirA*), whereas these genes are patchy among *Prochlorococcus* strains (Fig. S6) (Moore et al. 2002, Martiny et al. 2009, Scanlan 2009, Partensky and Garczarek, 2010, Berube et al., 2019)” (Lines 233-237)

Related references---

Moore LR, Post AF, Rocap G, Chisholm SW. 2002. Utilization of different nitrogen sources by the marine cyanobacteria *Prochlorococcus* and *Synechococcus*. *Limnol Oceanogr* 47:989-996.

Martiny AC, Kathuria S, Berube PM. 2009. Widespread metabolic potential for nitrite and nitrate assimilation among *Prochlorococcus* ecotypes. *Proc Natl Acad Sci U S A* 106:10787–10792.

Scanlan DJ, Ostrowski M, Mazard S, Dufresne A, Garczarek L, Hess WR, Post AF, Hagemann M, Paulsen I, Partensky F. 2009. Ecological genomics of marine picocyanobacteria. *Microbiol Mol Biol Rev* 73:249-99.

Partensky F, Garczarek L. 2010. *Prochlorococcus*: advantages and limits of minimalism. *Ann Rev Mar Sci* 2:305-31.

Berube PM, Rasmussen A, Braakman R, Stepanauskas R, Chisholm SW. 2019. Emergence of trait variability through the lens of nitrogen assimilation in *Prochlorococcus*. *eLife* 8.

- L. 323: why were only NO_3^- and NH_4^+ chosen to be part of the model (and not urea, for which data was also available?)

We have selected NO_3^- and NH_4^+ to be part of the model mainly to focus on delivering the most striking finding extracted from the data - the different affinity to chemical forms of inorganic nitrogen between two organisms. Also, inorganic nutrients are more commonly used in the ecological modeling and we have followed the previous similar studies focusing on inorganic nutrients as a starter (Tilman, 1977, Ward et al., 2013). Now we included these in the text (Lines 124-128)

Related references---

Tilman, D. 1977. Resource competition between plankton algae: An experimental and theoretical approach." *Ecology* 8: 338-348.

Ward, B. A., et al. (2013). "Iron, phosphorus, and nitrogen supply ratios define the biogeography of nitrogen fixation." *Limnology and Oceanography* 58(6): 2059-2075.

- L. 346-347: this is a bit unclear, maybe change to "For Fig. 3A, 4C, S5A, S5B, we varied the ratios of half-saturation constants, using $\text{NO}_3^-:\text{NH}_4^+$ resource ratios of 1:3, 2:3, 1:6 and 3:1, respectively"

The sentence was revised following the suggestion (Lines 450-452).

"For Fig. 4A, 5C, S5A, S5B, we varied the ratios of half-saturation constants, using $\text{NO}_3^-:\text{NH}_4^+$ resource ratios of 1:3, 2:3, 1:6 and 3:1, respectively."

- L. 348-354 This reads more like a discussion, and should to my opinion be included in the main text.

We appreciate the reviewer's comments. This paragraph is to explain equation formation, which appears first time in the method section. Because the method section is located after the discussion, we feel that this is best suited here. We spent time searching for places where this paragraph may fit, but we could not identify a place where this paragraph may be inserted without disturbing the flow of the main text. However, if the reviewer identifies a place for inserting this paragraph, we are happy to reconsider moving it.

- L. 396 "For N and P addition, data is based on Exp. M1-M3, where the initial concentrations of N were small": since the initial concentrations of N are used to select which data are represented on the figure, as well as for parametrizing the model (e.g. N quotas per cell), I believe that table S1 in ref 14 should be included in this manuscript as a supplemental table.

We have included Table S1 of Masuda et al., 2022 (previous ref. 14) as the new Table S2.

- L. 398 "Note that added NH_4^+ was depleted on the third day": again, this is an important information to interpret the results, and I believe that Fig S1 in ref 14 should be included in this manuscript as well. Or otherwise refer to Fig S2.

Thank you for your suggestion. We stated that

"The enriched NH_4^+ was depleted on the third day, leading to decreased populations in both organisms (Fig. 1, Fig. S2)." (Lines 100-101)

and included Fig. S1 of Masuda et al., 2022 (previous ref. 14) as the new Fig. S2.

- L.411-412 Please explain in the legend (and maybe in the main text as well) why a ratio of 1:3 was used as a reference.

We selected a value, which allows the coexistence of *Prochlorococcus* and *Synechococcus* with the optimized parameterization. We added this explanation in the caption of Fig. 4.

"The model output is based on the resource ratio of NO_3^- and NH_4^+ is 1:3 (for other ratios, see Fig. 3, S6), which allows the coexistence of these organisms with the optimized ratios."

- Fig S1 legend "Prochlorococcus prefers NH₄⁺ while, Synechococcus prefers NO₃⁻." From what I see, I would include urea in this sentence. In addition, I don't think that these incubations tested for preference, but rather that Prochlorococcus was more efficient in growing on NH₄⁺ and urea, and Synechococcus more efficient in growing on NO₃⁻ (and urea). Due to transcriptomic regulation, if both were present at the same time, I am not sure that Synechococcus would prefer NO₃ over NH₄.

We revised the sentence. Now it reads, "*Prochlorococcus* grows better with NH₄⁺ and urea while *Synechococcus* grows better with NO₃⁻."

Typos

- L. 130: Table S6 does not show K values. The authors probably meant Table S3. We revised the mistake, the current number is Table S4.

- L. 133-134 "On the other hand, K for NO₃⁻": this is probably a typo to be removed.

Sorry for the mistake. We completed the sentence.

"On the other hand, *K* for NO₃⁻ (*K*_{NO₃}) must be lower for *Synechococcus* to reproduce its continued high growth up to day 3 (Fig. 2B) and reversal of species NO₃⁻ affinities again results in a poor model fit to data (Fig. S5)." (Lines 136-138)

- L. 141: it is table S3, not S5.

Now we deleted the related sentence.

- L. 144: Table S4, not S5.

We revised the mistake accordingly. The current number is Table S5.

- L. 351 "and model the model": typo.

We revised the mistake. Now it reads; "Despite the simplicity, the model reproduces the population data (Fig. 2) under N limitation, which accounts for a large part of the *Prochlorococcus* and *Synechococcus* habitats." (Lines 457-458)

- L. 403 Points in figure 2 are huge (in Prochlorococcus, they are almost 10000 cells/mL wide). I would suggest to reduce point size.

Following the reviewer's comment, we have reduced the size of the point for Fig. 2, Fig. S2-4.

- L. 411-412 "The model output is based on the resource ratio of NO₃⁻ and NH₄⁺ is 1:3 (for other ratios, see Figure 3, S5)." There are 2 typos here (one extra "is", and Figure 3 is mentioned in its own legend!).

We thank the reviewer for spotting these. We revised the mistake. Now it reads: "The model output is based on the resource ratio of NO₃⁻ and NH₄⁺ is 1:3 (for other ratios, see Fig. 5, S6), which allows the coexistence of these organisms with the optimized ratios." (Lines 539-541)

- Table S2: typo in M1, day3, Prochlorococcus: (Cont = PO₄) < (NO₃ = Urea) < Urea. Typo for Fe1, Fe2, Fe3 for Day1 in Prochlorococcus and Synechococcus: includes NO₃ and NH₄ and Urea instead of Fe and PO₄.

We revised the mistake in new Table S3. "(Cont = PO₄) < (NO₃ = Urea) < Urea" in M1, day3, Prochlorococcus is replaced to "(Cont = PO₄) < (NO₃ = Urea) < NH₄", "Cont = PO₄ = NO₃ = NH₄ = Urea" in Fe1, Fe2, Fe3 day1 are replaced to "Cont = Fe = PO₄ = Fe+PO₄ = Fe+NO₃".

June 7, 2023

Dr. Takako Masuda
Kokuritsu Kenkyu Kaihatsu Hojin Suisan Kenkyu Kyoiku Kiko
Fisheries Resources Institute, Marine Environment Division
Shiogama
Japan

Re: Spectrum04000-22R1 (Coexistence of dominant marine phytoplankton sustained by nutrient specialization)

Dear Dr. Takako Masuda:

Thank you for addressing all reviewers comments and suggestions carefully. On data availability, you have written: "The data supporting this study are available on request from the corresponding author (TM)". Accordingly, to the ASM policies, all data should be available. Therefore, you will be request to change your statement and provide all data in open repository or unstructured database.

Your manuscript has been accepted, and I am forwarding it to the ASM Journals Department for publication. You will be notified when your proofs are ready to be viewed.

Sincerely,

Adriana Lopes dos Santos
Editor, Microbiology Spectrum
